# Retrograde signals control dynamic changes to the chromatin state at photosynthesis-associated loci

Marti Quevedo [1,2] ✉, Ivona Kubalová [1,4], Alexis Brun [1,4], Luis Cervela-Cardona [1,3], Elena Monte [2] & Åsa Strand [1] ✉

Retrograde signalling networks originating in the organelles dictate nuclear gene expression and are essential for control and regulation of cellular energy metabolism. We investigate whether such plastid retrograde signals control nuclear gene expression by altering the chromatin state during the establishment of photosynthetic function in response to light. An *Arabidopsis thaliana* cell culture provides the required temporal resolution to map four histone modifications during the greening process. We uncover sequential and distinct epigenetic reprogramming events where an epigenetic switch from a histone methylation to an acetylation at photosynthesis-associated loci is dependent on a plastid retrograde signal. The transcription factors VIVIPAROUS1/ABI3-LIKE (VAL1), RELATIVE OF EARLY FLOWERING 6 (REF6) and GOLDEN2-LIKE FACTOR1/2 (GLKs) are linked to the H3K27ac deposition at photosynthesis associated loci that precedes full activation of the photosynthesis genes. Our work demonstrates that retrograde signals play a role in the epigenetic reprogramming essential to the establishment of photosynthesis in plant cells.

Chloroplast biogenesis is a complex, highly regulated process where the photosynthetic components are encoded by both nuclear and plastid genomes. Thus, the establishment of photosynthesis is controlled by a delicate interplay between the two cellular compartments[1]. Chloroplast development proceeds in two regulatory phases[2,3], where the first phase is initiated and choreographed by nuclear events and referred to as anterograde control, whereas the second phase is dependent on plastid activity and retrograde signals[3]. The dependency of the expression of photosynthesis associated nuclear genes (*PhANGs)* on plastid activity provides a clear checkpoint which enables the plant to synchronize expression from the nuclear and chloroplast genomes during the greening process and seedling establishment[4,5]. The first phase of chloroplast development starts with light activation of the photoreceptors and associated transcription factors[6,7]. Transcriptional regulation is key to light response and

photomorphogenesis and is largely controlled by a specialized set of transcription factors including ELONGATED HYPOCOTYL 5 (HY5) and a family of PHYTOCHROME INTERACTING FACTORS (PIFs) each targeting numerous genes leading to major changes in nuclear transcription[8]. In addition, the first light response is also mediated by adjustments of the epigenome and the action of chromatin remodellers[8,9]. The fundamental unit of chromatin in eukaryotes is the nucleosome, which consists of a histone octamer wrapped by DNA in almost two turns. The histone proteins are small, globular, and positively charged, and with flexible N-terminal "tails" that protrude from the core structure. These "tails" act as a platform for reversible, chemical post-translational modifications (PTMs) that alter chromatin structure and function. Histone modifications have been shown to be modulated by the photoreceptors in response to the first hour of light[8,10,11] and activation of the *PhANGs* has been linked to histone

[1]Umeå Plant Science Centre, Department of Plant Physiology, Umeå University, Umeå SE-901 87, Sweden. [2]Centre for Research in Agricultural Genomics (CRAG) CSIC-IRTA-UAB-UB, Campus UAB, Bellaterra, Barcelona 08193, Spain. [3]Group of Genetics, Breeding and Biochemistry of Brassicas, Misión Biológica de Galicia, Spanish Council for Scientific Research (MBG-CSIC), Pontevedra 36143, Spain. [4]These authors contributed equally: Ivona Kubalová, Alexis Brun. ✉ e-mail: marti.quevedo@cragenomica.es; asa.strand@umu.se

acetylation in several plant species[12–15]. Histone acetylation on positively charged lysine residues leads to neutralization of charges that weakens the interactions between the histone and DNA, thus resulting in open chromatin that is more accessible to regulators[16]. The histone deacetylase HDA15 has been shown to be linked to light signaling pathways through regulating chlorophyll biosynthesis, photosynthesis gene expression and hypocotyl elongation[17,18].

The second phase of chloroplast development takes place after the initial anterograde signals, and it was demonstrated that a retrograde signal originating from the maturing plastids is essential for the complete activation of the photosynthesis-associated transcriptome[3]. This retrograde signal involves the plastid protein GENOMES UNCOUPLED 1 (GUN1) and the nuclear transcription factors GLKs[19]. Disturbing chloroplast biogenesis dramatically inhibits the final activation of *PhANG* expression[20] emphasizing the importance of the signalling system that synchronizes expression from the nuclear and plastid genomes. It is clear that retrograde signals are essential to the establishment of photosynthesis during the greening process but whether retrograde signals control nuclear gene expression by regulating the epigenome has so far not been addressed. In mammalian models, correct mitochondrial function was shown to be linked to the nuclear epigenetic landscape[21] but no mechanistic detail has so far been revealed.

We herein ask if control of the epigenome is closely connected to organellar activities by retrograde signals and specifically if plastid signals could trigger changes to the chromatin. To address this question, we explored the dynamic landscape of four histone modifications during the two regulatory phases driving the establishment of photosynthesis. We took advantage of an established homogenous and synchronous *Arabidopsis thaliana* cell culture[3,22] providing the temporal resolution required to decipher the regulatory mechanisms controlling the transition from a heterotrophic sink to a photoautotrophic source cell. We discovered a mechanism by which retrograde signals trigger a specific switch in histone modification at photosynthesis-associated loci. This epigenetic reprogramming is essential for the activation of *PhANGs* during the second phase of the process of chloroplast development and for the establishment of photosynthetic activity.

## Results

### The establishment of photosynthesis is a highly dynamic epigenetic process

Until now, it has proven difficult to separate the impact of plastid signals on chromatin regulation from the action of photoreceptors, as both pathways coincide during early seedling development. To reach the temporal resolution required to separate independent reprogramming events controlling the transition from a heterotrophic sink to a photoautotrophic source cell we took advantage of an established homogenous and synchronous *Arabidopsis thaliana* suspension cell culture[3,22] (Fig. 1a). We selected four histone marks to cover the main regulatory states[23], histone H3 lysine 4 tri-methylation (H3K4me3) and lysine 27 acetylation (H3K27ac) for active regions[24,25], histone H3 lysine 27 tri-methylation (H3K27me3) for repressed regions[26,27] and lysine 9 di-methylation (H3K9me2) for silenced repetitive regions[28] (Supplementary Fig. 1a). We sampled four time points starting with the dark-grown cell culture (Dark), followed by 1, 4 and 7 days growth in constant light (Day1, Day4 and Day7) (Fig. 1a). At Day7 photosynthetic functional chloroplasts have been established in the cell culture[3,22]. The ChIP-seq data for the histone marks was correlated with published RNA-seq data from the same time points[3]. As previously reported, H3K4me3 and H3K27ac associated with highly transcribed genes, H3K27me3 with genes with low transcriptional activity, and H3K9me2 associated with repressed transposable elements (Supplementary Fig. 1b,c). The chromatin landscape of the green cell culture on Day7 was compared to published ChIP-seq data from mature *Arabidopsis*

seedlings (Supplementary Data 1) to confirm that the two experimental systems are very similar (Fig. 1b, Supplementary Fig. 1d).

Next, we determined differentially enriched regions (DERs) for each histone mark at each time point. We curated a consensus peak set for all histone marks and time-points and calculated differential binding affinity between the PTM ChIP-seq data and their paired H3 control. This allowed normalization for nucleosome occupancy by comparing PTM signals with total H3 levels. Our differential analysis revealed three distinct phases at the chromatin level. The activation marks H3K27ac and H3K4me3 displayed similar patterns with very dynamic transitions from Dark to Day1 and secondly from Day4 to Day7 (Fig. 1c), following the described anterograde and retrograde controlled phases of transcription[3] and similar to what previously have been shown during the early response to light[8,10]. While no major differential expression was observed between Day1 and Day4[3], an intermediate epigenetic phase was observed characterized by a massive loss of H3K27me3 (Fig. 1c). Overall, we annotated DERs to more than 11500 loci emphasizing the suitability of our cell culture model to study dynamic chromatin transitions (Supplementary Data 2,3).

We further investigated the biological processes linked to the observed epigenetic changes. On Day1, early light signals depleted the activation marks H3K27ac and H3K4me3 from genes associated with growth in darkness (Fig. 1d, Supplementary Data 4), while the marks accumulated in genes encoding proteins promoting growth in the light, cell differentiation, cytoskeleton organization and photosynthesis, all terms linked to early light response and photomorphogenesis[29] (Fig. 1d, Supplementary Data 4). From Day1 to Day4 a loss of H3K27me3 was observed at genes involved in cell wall organization (Fig. 1d), which is consistent with previous reports on the effects of phytochrome on cellulose synthesis[30]. Additionally, we also observed that genes with depleted H3K27me3 were mostly coupled to molecular functions such as oxygen and tetrapyrrole binding (Supplementary Fig. 1e, Supplementary Data 4). The transition from Day4 to Day7 demonstrated activation of loci associated with photosynthesis and plastid organization, gene ontologies known to respond to the retrograde signal (Fig. 1d). Altogether, our chromatin roadmap covered the key regulatory stages during the establishment of photosynthesis and revealed sequential reprogramming steps including a loss of histone methylation followed by a late gain in histone acetylation.

### Photosynthesis loci switch from H3K27me3 to H3K27ac during the greening process

Activation of *PhANGs* has been linked to histone acetylation in several plant species[12–15]. However, previous studies focused on the early light response and regulation of histone marks via the action of phytochromes or compared dark-grown seedlings to completely mature green seedlings[8,10,17]. In contrast, our cell culture system allowed for detailed temporal profiling of histone mark deposition. We were able to monitor tight temporal changes in H3K27ac that were closely related to RNA expression levels (Supplementary Fig. 2) and revealed a significant subset of *PhANGs* gaining acetylation first at the stage during chloroplast development that correlated with the second induction of gene expression (Fig. 2a, b)[3]. In addition, we explored the overlap between genes losing the H3K27me3 mark on Day4 while gaining H3K27ac on Day7 (Fig. 2c, d). Despite the modest overlap, the resulting subset was highly enriched with photosynthesis genes (Fig. 2e). H3K27me3 depletion has previously been linked to tissue-specific gene activation during differentiation[31,32]. Our data from the Arabidopsis cell culture indicated that a significant group of photosynthesis genes is regulated by early repression and late acetylation (Supplementary Data 4).

To confirm the observed switch from H3K27me3 to H3K27ac *in planta* we followed the chromatin state of selected *PhANGs* during seedling de-etiolation, the transition of seedlings grown in the dark and shifted to light[33] We performed ChIP-qPCRs at 8 key *PhANG* loci

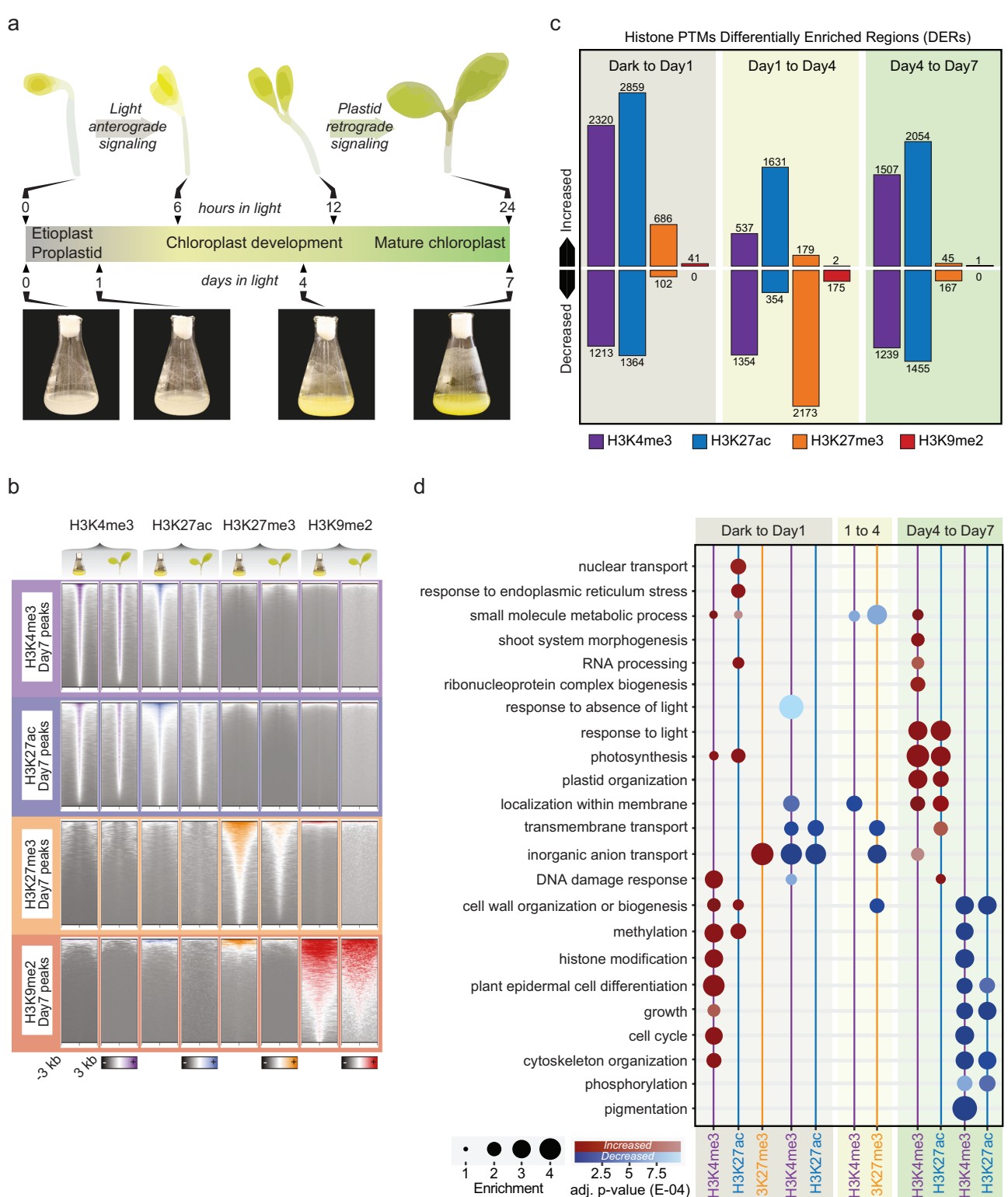

plus 2 separate sets of control genes (two primer pairs for each histone PTM, Control 1-4) along the four time-points during seedling de-etiolation (Dark, and 6, 12, and 24 h after light exposure) (Fig. 2f, Supplementary Fig. 3) and combined the normalized profiles in a meta-qPCR plot (Fig. 2g). We observed a fast H3K27ac deposition after 6 h of light but with a decline at 12 h, supporting a described maintenance mechanism for dynamic nucleosome PTMs[34,35]. At 12 hours light exposure H3K27me3 deposition at the *PhANG* loci is high and H3K27ac low which is symptomatic of the critical checkpoint of the chloroplast

development process at this time point[3]. In addition, compared to the homogenous and synchronized cell culture, cell-to-cell variation in seedlings at this critical developmental stage may impact the final readout in the seedlings. However, following 24 hours of light exposure, the chloroplasts have matured, and the retrograde signal has been triggered. At this time point the H3K27me3 mediated repression was reduced while H3K27ac gained, resulting in an increased expression of *PhANGs* (Fig. 2g). Thus, we hypothesize that retrograde signals play a key role in this final transition from H3K27me3 to H3K27ac

**Fig. 1 | Genome-wide mapping of histone post-translation modifications (PTMs) during the establishment of photosynthesis. a** Schematic representation of the establishment of photosynthesis in the *Arabidopsis* cell suspension culture. Cells were collected at four time points termed Day0 (Dark), Day1, Day4 and Day7 following exposure to constant light during the greening process. **b** Heatmaps displaying the overlap between Day7 *Arabidopsis* cell culture and published *Arabidopsis* green seedling data of total chromatin Immunoprecipitation (ChIP)-sequencing(seq) peaks for each histone PTM. The ChIP signal of each histone PTM is ranked according to all peaks identified in Day7 cell culture for that histone PTM. Each column group displays histone PTM ChIP-seq heatmaps of Day7 cell culture or 10 days after germination (DAG) seedlings. Heatmaps are arranged in rows for each set of called cell culture peaks of different histone PTM. At each heatmap, lines correspond to peaks which are sorted based on ChIP-seq signal and displayed in a 6 kb window centred at the peak summit. The same scale is used for cell culture and seedling heatmaps. **c** Bar plot representing Differentially Enriched Histone PTM Regions (DERs at each transition across the experiment. The selecting criteria were FDR < = 0.05; Fold > 2. **d** Gene Ontology (GO) enrichment analysis of the genes annotated to histone PTM DERs across the experiment. The size of the circles indicates the enrichment (% found vs % expected). The red colour is used for increased DERs and blue for decreased DERs, the colour intensity indicates the significance assessed using a two-sided Wilcoxon rank-sum test with Bonferroni correction applied for multiple comparisons. The complete tables of enriched GO terms are presented in Supplementary Data 4.

during the second phase of chloroplast biogenesis and establishment of photosynthesis.

## Retrograde signal(s) control the chromatin state at *PhANG* loci

To test the involvement of a retrograde signal in the switch in histone modification observed during the greening process, we used lincomycin, an inhibitor of plastid protein translation that blocks chloroplast biogenesis[36] (Fig. 3a, d and Supplementary Fig. 4a, b). Lincomycin treatment locks the seedlings in the developmental state prevailing in darkness by stabilizing GUN1 and repressing *PhANGs* by blocking the second phase of transcriptional activation of *PhANGs*[19] (Supplementary Fig. 4a,b). When lincomycin was applied to the cell culture the late Day4-Day7 gain in acetylation was lost, and high levels of H3K27me3 were maintained at the *PhANG* loci (Fig. 3b, c). A silenced chromatin state at the *PhANG* loci was also observed when *Arabidopsis* seedlings were grown on lincomycin (Fig. 3e, f). In addition to the lincomycin treatment of wild-type seedlings we also grew the *gun1* mutant on lincomycin. In the *gun1* mutant the suppressive signal prevailing in the dark to block premature expression of *PhANGs* is absent. In the *gun1* mutant *PhANG* expression is elevated compared to wild-type, the deposition of H3K27Ac is increased following growth on lincomycin and the deposition of H3K27me3 decreased compared to wild type (Fig. 4a, b, Supplementary. Figure S11). These observations support a mechanism by which a retrograde signal involving GUN1, modulates a switch in chromatin modification at the photosynthesis loci.

## Identification of regulators of the chromatin state at *PhANG* loci

We hypothesize that regulators of the establishment of photosynthesis influence the chromatin state at the *PhANG* loci. To identify putative regulators of the chromatin landscape we performed de novo motif analysis to discover *cis*-elements differentially enriched at histone PTM DERs along the course of the greening process[37]. The transcription factors GLKs, REF6 and VAL1 were identified from the motif prediction analysis, (Fig. 5a, Supplementary Fig. 5). GLK2 and its paralog GLK1 are members of the GARP family of Myb transcription factors with a well-established role during chloroplast development and regulation of photosynthesis across a range of plant species[38–40]. GLKs act downstream of plastid retrograde signals and are targets of GUN1-mediated signalling[41] and were recently connected to histone acetylation[14]. REF6, also known as Jumonji domain-containing protein 12 (JMJ12), is a C2H2 zinc finger transcription factor with H3K27me2/3 demethylase activity[42,43] that counteracts Polycomb in a wide range of processes, including leaf and flower development[44] and thermomorphogenesis in cooperation with PIF4[45]. Importantly, a REF6-dependent H3K27me3-depleted state has been shown to facilitate gene activation during germination[46]. The *ref6* mutant was also shown to have impaired seed germination, and impaired expression of photosynthesis genes during senescence[47,48]. VAL1 is a well-described transcriptional repressor, that together with VAL2, recruits the Polycomb repressive complex 2 (PRC2) for genome-wide Polycomb-mediated silencing in *Arabidopsis*[49]. Due to its essential role in recruiting components of the chromatin silencing machinery, VAL1 is linked to many developmental processes, including seed dormancy and floral induction[50–52]. Thus, GLK2, REF6 and VAL1 were selected for further analysis of their potential involvement in the switch in chromatin state at the photosynthesis loci during the greening process.

The expression profiles of *GLK2, REF6* and *VAL1* match the major chromatin events during the establishment of photosynthesis. *GLK2* showed increased expression during the second phase, *REF6* at both early and late acetylation events and *VAL1* showed decreased expression levels during the H3K27me3 removal phase (Fig. 5b). Available ChIP-seq data for these transcription factors (Supplementary Data. 1) was used to compare the occupancy of H3K27me3 and H3K27ac in our experiment, with signals for VAL1, REF6 and GLK2, respectively at our subsets of interest (Fig. 5c, d, Supplementary Fig. 6, 7a). VAL1 occupancy was found at the clusters of photosynthesis genes that were initially enriched for H3K27me3 (Fig. 5d, Supplementary Fig. 8). GLK2 and REF6 occupancy was found at loci acetylated during the Day4-Day7 transition, with GLK2 preferentially binding around the histone PTM regions, while REF6 centred at or spanned along the peaks (Fig. 5d). From the pool of genes gaining acetylation from Day4 to Day7, we observed that a group of genes referred to as "Profile5" (Supplementary Fig. 2) comprised a small proportion of *PhANGs* that were acetylated exclusively at Day7, but with an absence of H3K27me3 and low H3 density from a start (Supplementary Fig. 6b,c). This permissive chromatin context was accompanied mostly by GLK2 binding. On the other hand, most VAL1 signal was observed at the "overlap" subset, the subset that is subject to a switch from 27me3 to 27ac during the greening process (Supplementary Fig. 6a,b). The "overlap" subset was investigated further, revealing VAL1 occupancy to be particularly found at 2 clusters (defined by VAL1 binding) (Fig. 5c, Supplementary Fig. 8a–d). These gene clusters contained several *PhANGs* (Supplementary Fig. 6e, 8b). Notably, a prominent GLK2 signal was also associated around the centre of the "overlap" regions. Additionally, as these loci transition from a repressive H3K27me3 state to a more acetylated permissive chromatin context, they also displayed a reduction of H3 occupancy (Supplementary Fig. 6c, 8d). These findings strongly support the idea that *PhANGs* undergo a nucleosome remodelling process granting these regions increased accessibility, possibly by a tug-of-war between VAL1 and GLKs.

To directly investigate the involvement of VAL1, REF6 and GLK1/2 in the greening process we retrieved insertion lines and overexpression lines for these factors (Supplementary Fig. 9). We observed a delay in the greening process for the *glk1;glk2* double mutant, similar to previous reports[38–40] with reduced accumulation of chlorophyll following 10 hours of light exposure. In addition, VAL1 over expression (VAL-ox) resulted in a pale phenotype (Fig. 5e, f, Supplementary Fig. 10a). In addition to the *ref6* mutant we collected two other mutants described to be affected in H3K27me3 deposition, the Histone Demethylases, Early Flowering 6 (ELF6) and CURLY LEAF (CLF), a histone methyltransferase of Polycomb Repressive Complex 2[44,53,54] and tested those for greening phenotype during the deetiolation process (Supplementary Fig. 11a–b). Interestingly only the

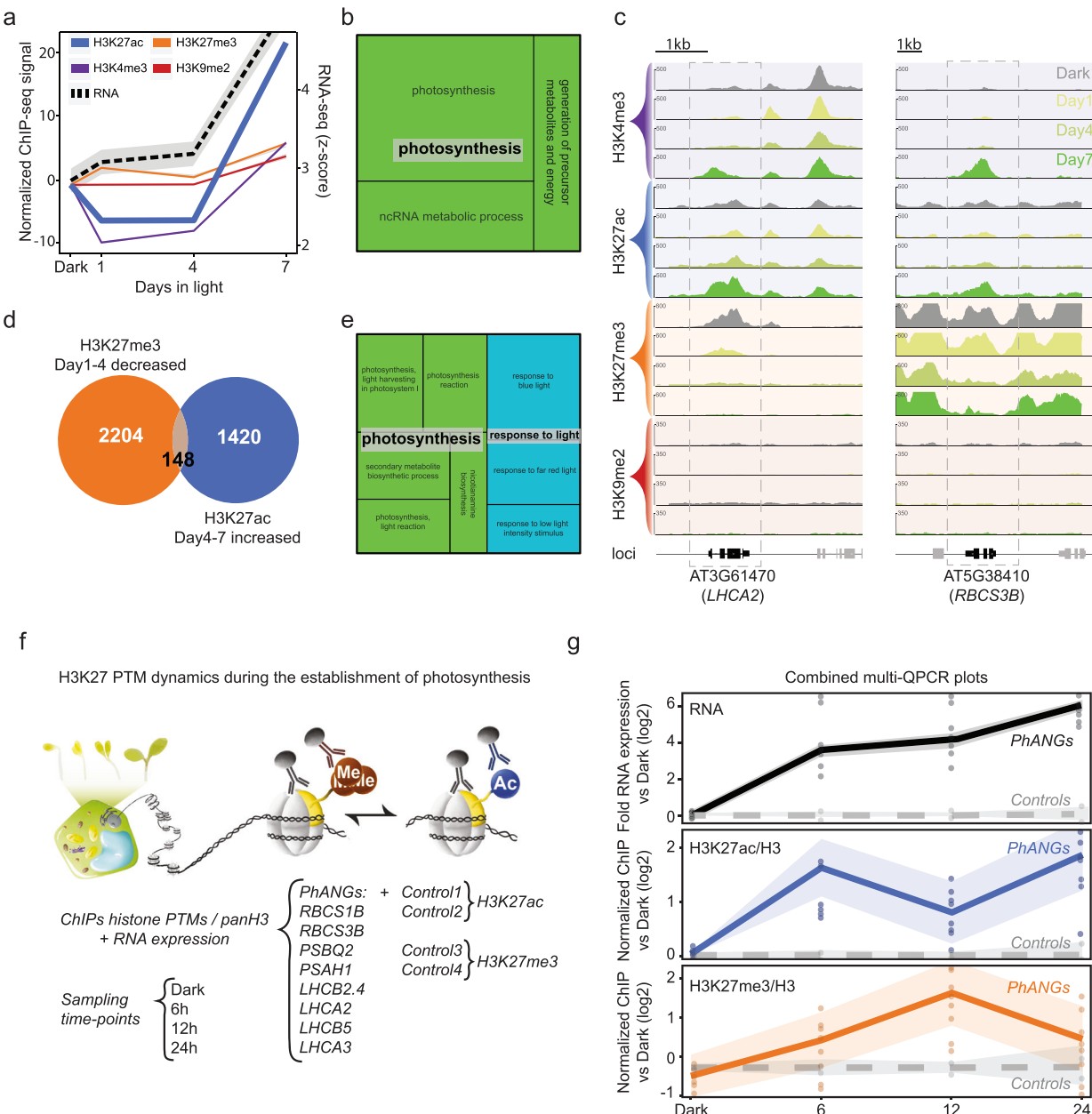

**Fig. 2 | Dynamic changes to the histone PTMs at *Photosynthesis-Associated Nuclear Genes* (*PhANGs*). a** Activation of *PhANGs* has been linked to a rapid histone acetylation in response to light linked to the action of photoreceptors in several plant species. Genomic regions identified that are specifically enriched for H3K27ac first at Day7 (Profile5, Supplementary Fig. 2). An average of the ChIP-seq signal for all histone PTMs, normalized to the Dark time-point, and RNA levels for those specific regions are displayed. **b** GO ontology terms found enriched in genes demonstrating a late H3K27ac deposition (Day7) using Tree Map adapted from REVIGO. **c** ChIP-seq visualization tracks of histone PTMs occupancy across the experiment at two example *PhANG* loci. A normalized ChIP-seq signal is indicated on the y-axis. A scale bar is indicated. **d** Venn diagram displaying genes that undergo both a significant decrease in H3K27me3 from Day 1 to Day 4 and a significant increase in H3K27ac from Day 4 to Day 7. **e** GO ontology terms found enriched at genes from the overlap displayed in Fig. 2d using Tree Map adapted from REVIGO. **f** Schematic representation of the establishment of photosynthesis during seedling de-etiolation depicting the two-phase model. Experimental set-up to study H3K27ac/H3K27me3 dynamics at *PhANG* loci during seedling de-etiolation. Samples were collected at four key time points (Dark and 6, 12, and 24 hours following exposure to constant light) in three independent biological replicates. H3K27ac, H3K27me3 and H3 ChIPs were processed together with an input sample from each condition by quantitative-polymerase chain reaction (qPCR). Eight primer pairs targeting different *PhANG* loci were evaluated in addition to two control genes for each histone PTM (Individual locus evaluation found in Supplementary Fig. 2). **g** Metaplot display of both normalized RNA, and ChIP-qPCR values of H3K27ac/H3 and H3K27me3/H3 at 8 *PhANG* loci versus their relative 2 control loci. The shaded area represents ± SEM of three independent biological replicates. Data is integrated from the individual targets presented in Supplementary Fig. 3a, b.

*ref6* mutants demonstrated significantly impaired greening compared to wild type. Exploring available H3K27me3 ChIP-seq datasets performed in *ref6* background plants revealed key chlorophyll biosynthesis genes *PORA*, *PORC* and *HEMA3* to have both REF6 binding and display H3K27me3 hypermethylation in the *ref6* mutant background (Supplementary Fig. 7b). Moreover, with newly generated REF6 overexpressing lines (REF6-ox, Supplementary Fig. 9) we observed REF6 overexpression to induce some *PhANG* expression in the dark and

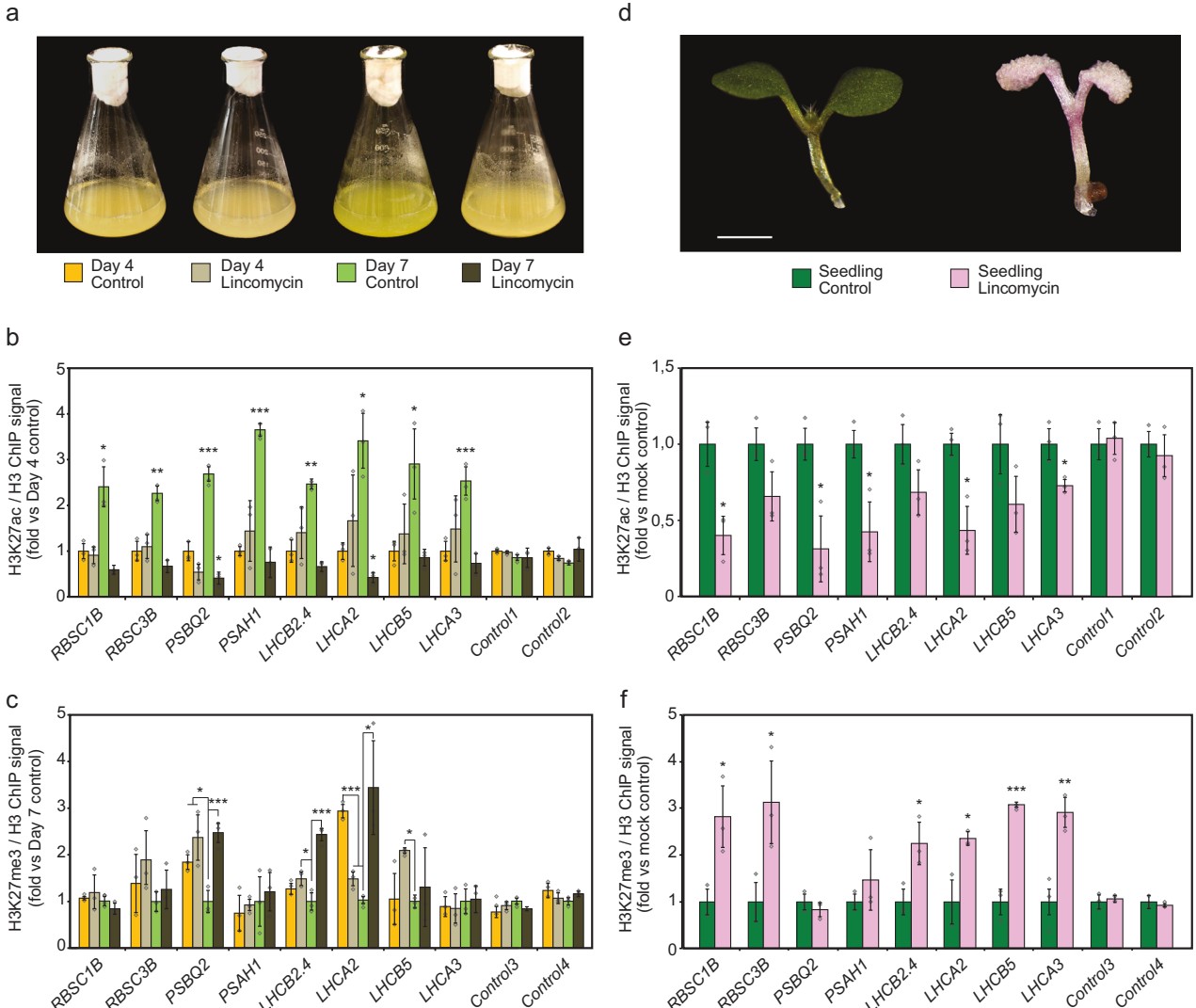

**Fig. 3 | Chloroplast retrograde signals influence chromatin dynamics at PhANG loci.** **a** Representative pictures of Day4 and Day7 *Arabidopsis* cell suspension culture treated with/without the chloroplast protein synthesis inhibitor lincomycin, the conditions used in triplicate for b and c panels. **b** ChIP-qPCR values for H3K27ac/H3 ratio at eight *PhANG* loci plus two control regions. The values were normalized by control region signal and to the Day4 untreated control condition. **c** ChIP-qPCR values for H3K27me3/H3 ratio at eight *PhANG* loci plus two control regions. The values were normalized by control region signal and to the Day4 untreated control condition. **d** Representative pictures of 7-day-old Col-0 *Arabidopsis* seedlings treated with/without lincomycin, the conditions used in triplicate

for e and f panels. **e** ChIP-qPCR values for H3K27ac/H3 ratio at eight *PhANG* loci plus two control regions. The values were normalized by control region signal and to the untreated seedling control condition. **f** ChIP-qPCR values for H3K27me3/H3 ratio at eight *PhANG* loci plus two control regions. The values were normalized by control region signal and to the seedling control condition. Data are presented as mean values from 3 independent biological replicates +/- SEM. Three independent biological replicates were used, and significance was assessed by Two-sample t-test with unequal variances (Welch's t-test) (*, $p < 0.05$; ** $0.05 > p > 0.001$; ***, $p < 0.001$). The individual data points are indicated by circles. Expression levels were quantified in the same samples and data presented in Supplementary Fig.4.

following growth on lincomycin (Supplementary Fig 11c-d). Thus, we focused our further studies on REF6. Attenuated expression of the *PhANGs* following 10 hours of light exposure was observed for the *glk1;glk2, ref6* and VAL1-ox lines (Supplementary Fig. 10). At the chromatin level, both the absence of GLKs, REF6 or the overexpression of VAL1 presented less H3K27ac deposition compared to the wild-type control seedlings following 10 hours light exposure (Fig. 5g). These results strongly indicate a role for VAL1, REF6 and GLKs in the regulation of *PhANGs* at the chromatin level during the establishment of photosynthesis.

The role of VAL1 during the establishment of photosynthesis could be explained by direct binding of VAL1 at the *PhANG* loci and the subsequent recruitment of the repression machinery PRC2 (Supplemental Figs. 6d, 7 and 8b). Besides the possible competition between

the repressive action of VAL1 at *PhANGs* and their activation by GLKs, we discovered a direct connection between VAL1 and GLK1, as published VAL1 ChIP-seq data revealed a prominent VAL1 peak downstream of the *GLK1* locus (Fig. 6a). No peak was found at the *GLK2* locus. In addition, *val1* and VAL1-ox (Supplementary Fig. 9h-k) showed increased and reduced *GLK1* expression levels, respectively, indicating that VAL1 might act as a repressor of GLK1 in the dark (Fig. 6b, c). To confirm the genetic interaction between VAL1 and the GLKs we created the *glk1;glk2;val1* triple mutant. Further support for the role of VAL1 controlling *GLK1* expression was shown by the higher expression of *GLK1* and *PhANGs* in the *glk1;glk2;val1* triple mutants compared to the *glk1;glk2* mutant (Fig. 6d, e, Supplementary Fig. 12a). *GLK1* is still expressed in *glk1*, albeit at lower levels compared to wild type (Fig. 6d)[38]. In the triple mutants, when the repressive action of VAL1 is

a

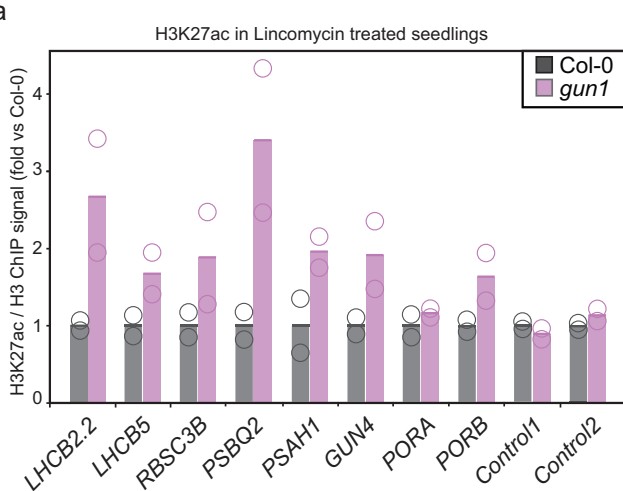

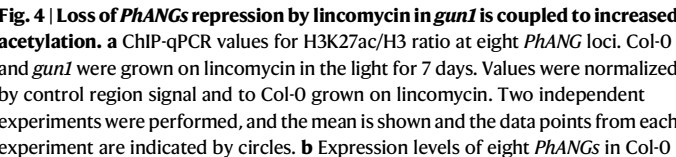

b

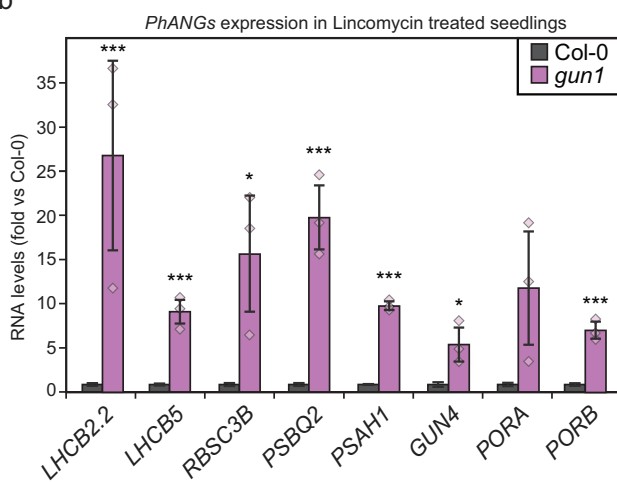

**Fig. 4 | Loss of *PhANGs* repression by lincomycin in *gun1* is coupled to increased acetylation. a** ChIP-qPCR values for H3K27ac/H3 ratio at eight *PhANG* loci. Col-0 and *gun1* were grown on lincomycin in the light for 7 days. Values were normalized by control region signal and to Col-0 grown on lincomycin. Two independent experiments were performed, and the mean is shown and the data points from each experiment are indicated by circles. **b** Expression levels of eight *PhANGs* in Col-0

and *gun1* grown in lincomycin for 7 days. Values were normalized to Col-0 and two reference genes (*UBC, EF1*) were used for normalization. Data are presented as mean values from 3 independent biological replicates +/- SEM. Three independent biological replicates were used, and significance was assessed by Two-sample t-test with unequal variances (Welch's t-test) (*, p < 0.05; ***, p < 0.001). The individual data points are indicated by circles.

removed, some *GLK1* expression is recovered resulting in higher expression levels of the GLK target genes (Fig. 6e). Thus, taken together these results present VAL1 as a direct regulator of the critical transcription factor, GLK1 and consequently its downstream target genes.

### *GLK2* over-expression compensates for the lack of a positive feedback signal from the plastids and activates *PhANGs* under conditions when chromatin is silenced

We have shown that plants with immature chloroplasts repress *PhANGs* at the chromatin level by maintaining the repressive H3K27me3 state (Fig. 3, Supplemental Fig. 4). However, the *gun1* mutant showed increased H3K27Ac deposition (Fig. 4). In addition, published data and our results indicate that GLK1/2 acts as a transcriptional activator of photosynthesis associated genes (Fig. 5). To further investigate if the action of GLKs is directly connected to the deposition of the H3K27ac at the *PhANG* loci, we retrieved a line overexpressing *GLK2* (GLK2-ox)[55] and generated a REF6 over expressor line (REF6-ox, Supplementary Fig. 9) and tested these lines under the lincomycin condition that maintains the silenced chromatin state (Figs. 3, 7a). The *GLK2* locus is not subject to the H3K27me3 to H3K27ac switch. In the GLK2-ox and REF6-ox lines, an increased deposition of H3K27ac was observed at the *PhANG* loci (Fig. 7a, Supplementary Fig. 11e). For the GLK2-ox line the increased deposition of H3K27ac was correlated with increased *PhANG* expression (Fig. 7b) suggesting a possible pioneer function for the GLKs. Pioneer transcription factors can unlock the chromatin and access their cognate binding motifs in the nucleosome[56–59]. The panH3 analysis from our ChIP-seq data show high H3 density during dark and following 1 day light exposure. The H3 density is reduced at the GLK targets from day4 and day7 and showed more accessible (less nucleosome dense) chromatin (Supplementary Fig. 6 and 8). Overall, overexpression of *GLK2* was able to rescue the silenced chromatin state under conditions of dysfunctional chloroplasts leading to increased acetylation and transcription at the photosynthesis-associated loci. The phenotype of GLK2-ox was similar to the phenotype of the *gun1* mutant where the retrograde signal suppressing *PhANG* expression in the dark is absent (Figs. 4, 7).

## Discussion

Chromatin acts as a signal integrator of diverse metabolic pathways[60], but how global metabolic fluctuations control specific and regional chromatin modifications is unknown. We have shown that plant biology provides an excellent model for these studies where a dynamic epigenetic landscape drives the establishment of photosynthesis leading to a transition from a heterotrophic sink to a photoautotrophic source cell. We provide a detailed epigenetic roadmap of the two regulatory phases driving the establishment of photosynthesis. We demonstrate that dynamic metabolic transitions are linked to changes to the histone code, that likely alter chromatin structure and DNA accessibility, giving rise to new cellular activities such as photosynthesis. We further demonstrated that the epigenome is closely connected to organellar activities by retrograde signals, and that a GUN1-mediated plastid signal is required for the specific changes to the chromatin state that permit full activation of photosynthesis genes and consequently the establishment of photosynthetic activity (Fig. 7c).

The process of chloroplast development depends upon two sequential and distinct epigenetic reprogramming events with a switch from histone methylation to acetylation at the *PhANG* loci. This reprogramming is directly correlated with the induction of photosynthetic gene expression (Fig. 7c). Possibly different kinetics are triggered by the early light signal, with de-methylation requiring more time, or even cell cycle progression. Our model suggests that the H3K27me3-mediated repressive mechanism needs to be overcome by a strong developmental cue, the retrograde signal (Fig. 3). Under lincomycin treatment in the REF6-ox line and the *gun1* mutant, we did observe significant removal of this mark compared to wild type at *PhANG* loci and a significant increase in H3K27ac (Fig. 4, Supplementary Fig. 11e,f). It has been reported that slow chromatin dynamics mediated by Polycomb act as a noise cancellation mechanism to avoid spontaneous activation of developmental programs[61] which is in line with the developmental checkpoint associated with the retrograde signal allowing coordination of the activities of the nuclear and plastid genomes[3].

We identified VAL1, REF6, and GLK2 as potential chromatin regulators acting at the sequential stages during the establishment of

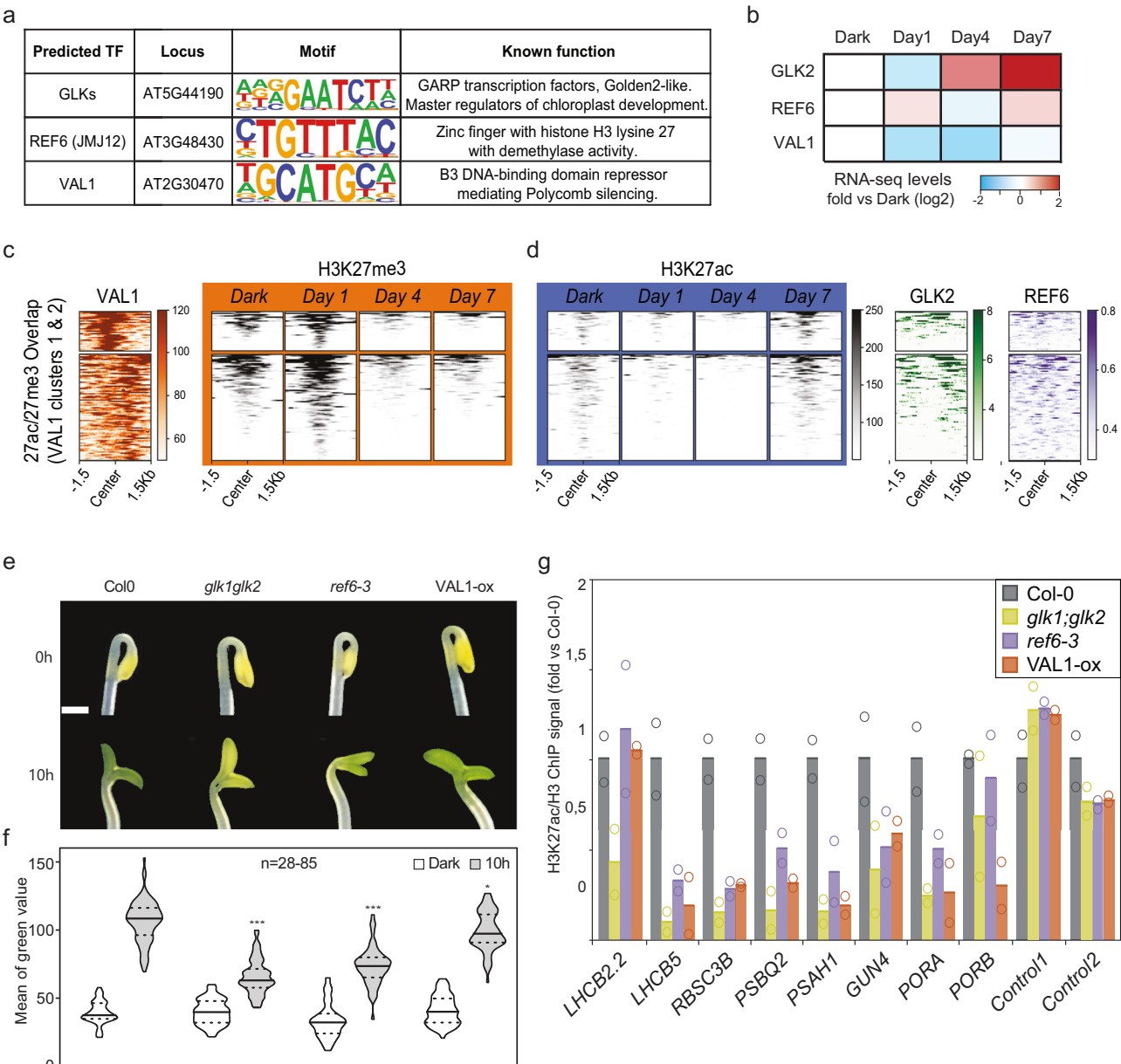

**Fig. 5 | Transcription factors involved in chromatin dynamics during the establishment of photosynthesis. a** Transcription factors identified from the HOMER Motif Analysis Algorithm of the temporal DERs or profile clusters. **b** Expression profiles of selected transcription factors during the establishment of photosynthesis in *Arabidopsis* cell culture. RNA-seq data was retrieved from Dubreuil et al. 2018. **c**, **d** Heatmaps of published ChIP-seq data for VAL1, GLK2 and REF6, along with histone PTM signals from this study, at the"overlap" subset (Fig. 2d)."Overlap" regions are ranked and clustered by the VAL1 peak signal (Supplementary Fig. 8). Each row represents one DER peak with a ± 1.5 kb of window centred to each DER region middle point. Scale bars represent ChIP-seq signal. **e** Representative photographs of 3-day-old dark-grown seedlings following 10 h of exposure to light. Scale bar 0.5 mm. **f** Quantification of green levels after image processing of seedling from RGB channels. Significance was assessed in n > 25 by One-way ANOVA test followed by Dunnett post-hoc test for comparison (*, $p < 0,05$; ***, $p < 0,001$). **g** ChIP-qPCR values for H3K27ac/H3 ratio at eight *PhANG* loci of 3-day-old dark-grown seedlings following 10 h of exposure to constant light. Values were normalized by control region signal and to the Col-0 control line. The mean of two independent experiments is shown and the data points from each experiment are indicated by circles.

photosynthesis. In our bioinformatic analysis the VAL1 binding motif was identified in regions losing the H3K27me3 mark before entering the second acetylation phase (Fig. 5). In our de-etiolation experiments, overexpression lines of VAL1 (VAL1-ox) showed impaired acetylation of *PhANG* loci, attenuated expression of *PhANGs* and delayed accumulation of chlorophyll (Fig. 5). VAL1 has been extensively characterized as a platform for recruitment of epigenetic repressors, including Polycomb Repressive Complex[49]. In a recent study, phytochrome B was found to recruit the PHD finger protein VIN3-LIKE1/VERNALIZATION 5

(VIL1/VRN5), a component of Polycomb Repressive Complex 2 (PRC2) in a light-dependent manner[62]. Light stabilizes the VIL1 protein, and light-activated phyB interacts with VIL1 to mediate chromatin remodeling through the deposition of H3K27me3. The deposition of H3K27me3 mediates a repression of growth-promoting genes and the *vil1* mutant displayed longer hypocotyls when grown in light compared to wild type[62]. Unlike VIL1, VAL1, and VAL2 are DNA-binding proteins interacting with the RY motif[63]. Interestingly, the RY motif was found to be one of the elements coupled to the G-box elements

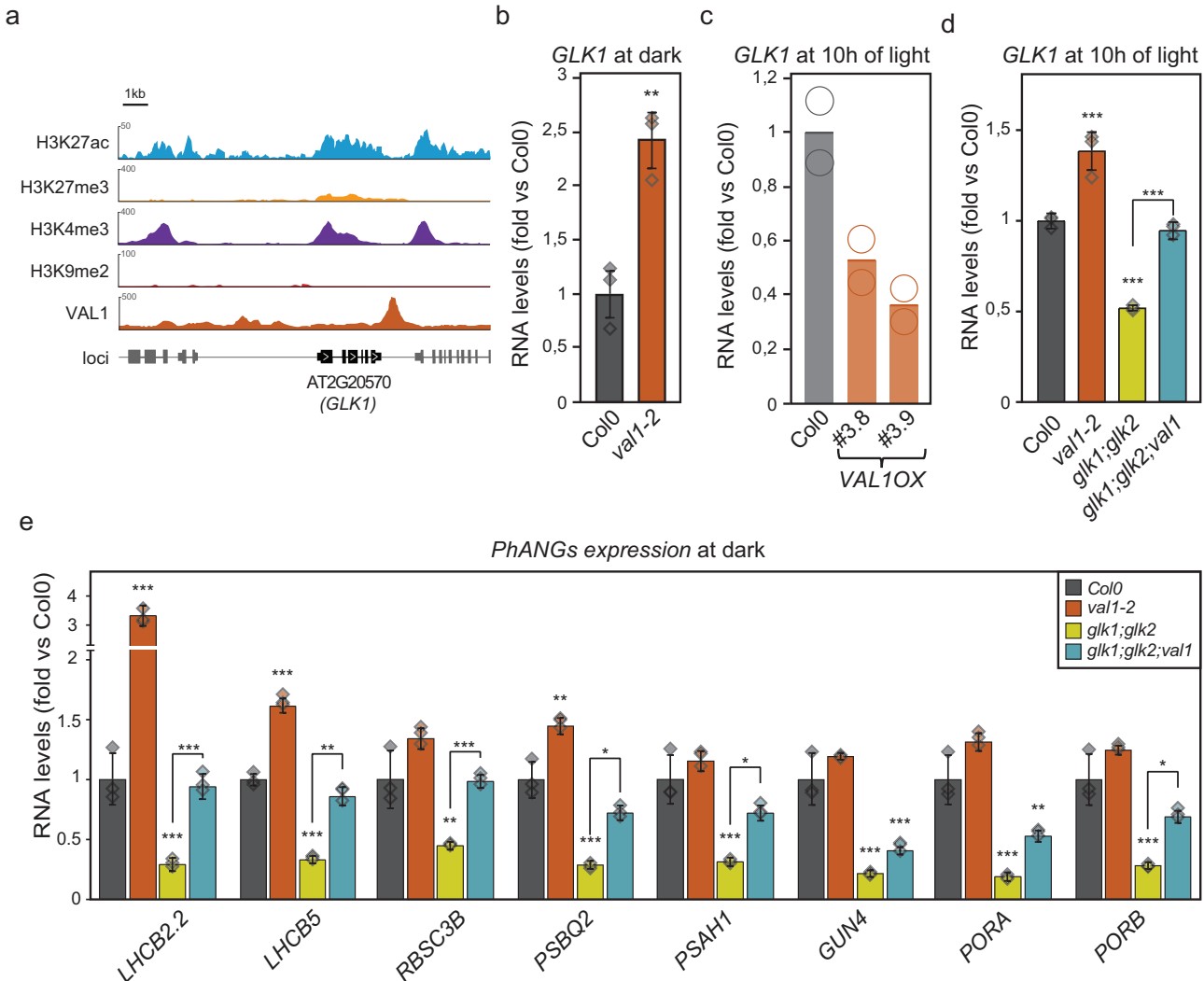

**Fig. 6 | VAL1 targets *GLK1* during the greening process. a** ChIP-seq visualization tracks at the GLK1 locus of histone PTMs and VAL1 occupancy from published seedling data (Supplementary Data 1). The normalized signal is indicated on the y-axis. The scale bar is 1 kb. **b** Expression levels of *GLK1* in Col-0 and *val1-2* mutant seedlings grown in darkness for 3 days. Values normalized to Col-0. Two reference genes (*UBC, EF1*) were used for normalization. **c** Expression levels of *GLK1* in Col-0 and VAL1-ox lines in 3-day-old dark-grown seedlings following 10 hours of exposure to light. Values were normalized to Col-0 and two reference genes (*UBC, EF1*) were used for normalization. The mean of two independent experiments is shown and the data points from each experiment indicated by circles. **d** Expression levels of *GLK1* in Col-0, *glk1;glk2, val1-2* and *glk1;glk2;val1* mutant seedlings grown in darkness for 3 days and transferred to light for 10 h. Values normalized to Col-0. Two reference genes (*UBC, EF1*) were used for normalization. **e** Expression levels of *PhANGs* in Col-0, *glk1;glk2, val1-2* and *glk1;glk2;val1* mutant seedlings grown in darkness for 4 days. Values normalized to Col-0. Two reference genes (*UBC, EF1*) were used for normalization. For panels **b,d,e** data are presented as mean values from 3 independent biological replicates +/- SEM. Three independent biological replicates were used, and significance was assessed by Two-sample t-test with unequal variances (Welch's t-test) (*, $p < 0.05$; ** $0.05 > p > 0.001$; ***, $p < 0.001$). The individual data points are indicated by diamonds.

where phyB and PIFs bind[64]. We identified VAL1 as a key regulator during the greening process, and possibly VAL1, like VIL1, interacts with phyB to give rise to the light-dependent regulation of PRC2 activity.

In contrast to VAL1, over-expression of GLK2 resulted in an increased deposition of H3K27ac that was correlated with increased *PhANG* expression (Fig. 7a, b). Under the lincomycin conditions when H3K27me3 is maintained (Fig. 3) and normal chloroplast development is inhibited, GLK2-ox compensates for the lack of normal biogenic retrograde signaling and activates expression of *PhANGs*. Our results support the conclusion that GLKs act downstream of plastid retrograde signals[41] and are linked to histone acetylation[14]. In addition, the ability of GLK2 to bypass chromatin silencing suggests a possible pioneer function for GLK2. Nucleosomes are refractory for most

transcription factor binding, however, a special class of transcription factors, termed pioneer transcription factors, can access their cognate binding motifs in the nucleosome[56–59]. These pioneer transcription factors play important roles in a range of developmental processes. For example, the mammalian pioneer transcription factor FoxA reprograms fibroblast to hepatocytes[58] while the Oct4, Klf4 and Sox2 pioneer transcription factors reprogram fibroblasts to induced pluripotent stem cells[56]. In plant systems only one transcription factor with pioneering activity has so far been described, LFY, which promotes the flowering process in plants[65]. Our model suggests that a major demethylation event precedes GLK action, which is in line with a recent report demonstrating that the pre-existing epigenetic landscape helps tune posterior pioneer activity[66]. A recent study has pointed to REF6 as a key facilitator of gene activation during

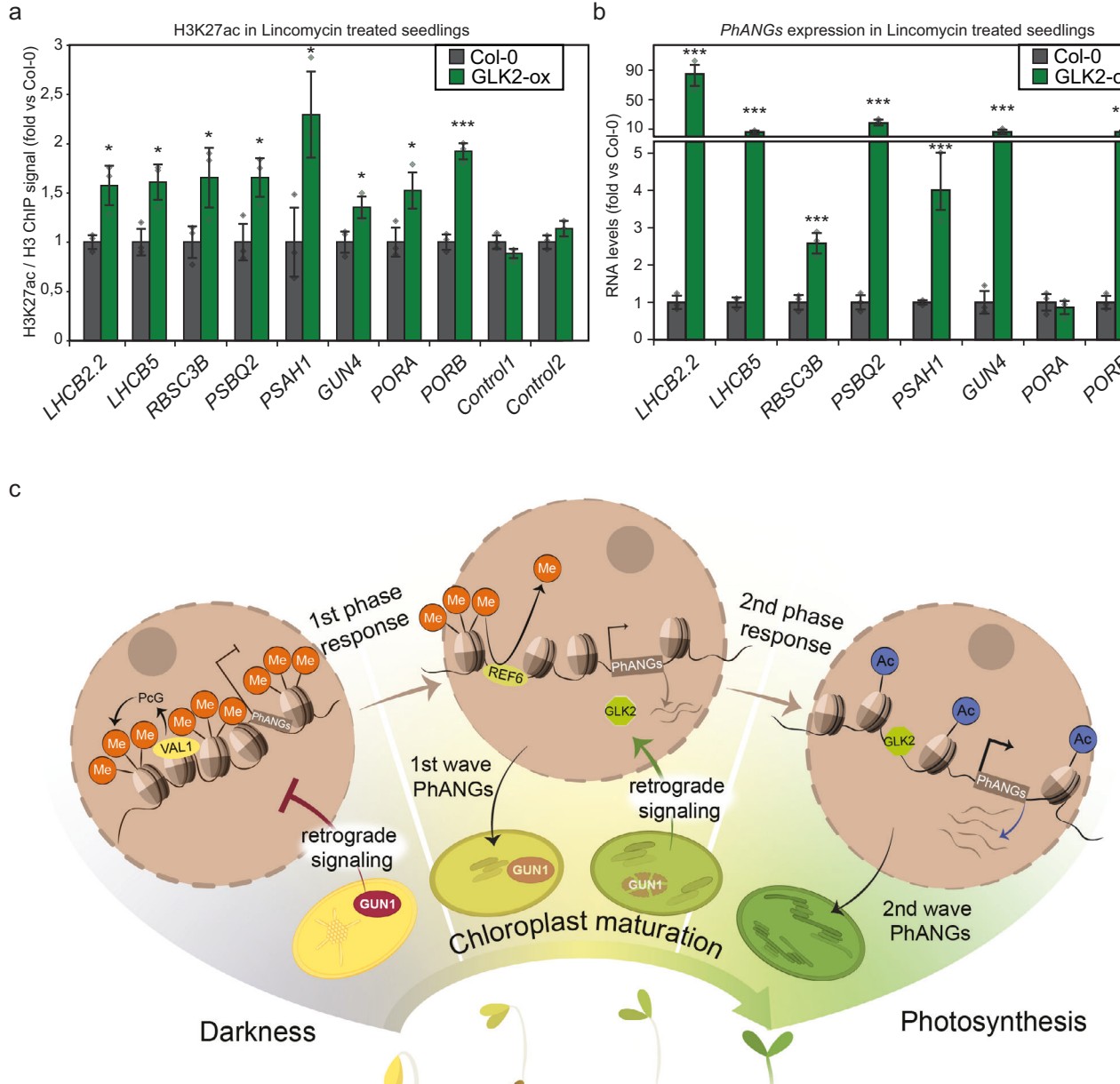

**Fig. 7 | Overexpression of GLK2 increases chromatin acetylation at *PhANG* loci and rescues *PhANGs* expression during growth on lincomycin. a** ChIP-qPCR values for H3K27ac/H3 ratio at eight *PhANG* loci. Col-0 and GLK2-ox were grown on lincomycin in constant light for 7 days. Values were normalized to Col-0 grown on lincomycin. **b** Expression levels of eight *PhANGs* in Col-0 and *GLK2* overexpression lines grown on lincomycin for 7 days and constant light. Values were normalized to Col-0 and two reference genes (*UBC, EF1*) were used for normalization. For panels a,b data are presented as mean values from 3 independent biological replicates +/- SEM. Three independent biological replicates were used, and significance was assessed by Two-sample t-test with unequal variances (Welch's t-test) (*, p < 0.05; ** 0.05 > p > 0.001; ***, p < 0.001). The individual data points are indicated by circles.

**c** Working model of how retrograde signals control dynamic changes to the plant epigenome. In the dark the *PhANG* loci are silenced by the repressive histone mark H3K27me3. VAL1, together with VAL2, recruits the Polycomb repressive complex 2 (PRC2) for silencing by depositing H3K27me3 and *PhANG* expression is repressed. The H3K27me3-mediated repressive mechanism at the *PhANGs* loci needs to be overcome by a strong developmental cue, the retrograde signal. Following the removal of the H3K27me3 mark possibly involving REF6, acetylation at the *PhANG* loci occurs and *PhANG* expression is induced. Possibly GLK2 plays an important role during the switch from H3K27me3 to H3K27ac in the epigenetic reprogramming that is essential to the establishment of photosynthesis in plant cells.

germination by setting a H3K27me3-depeleted state[46]. Here we show that REF6-mediated demethylation activity plays a role during the establishment of photosynthesis, probably acting before the action of GLKs. Interestingly, our data indicates that VAL1 acts as a repressor of specifically *GLK1* in the dark (Fig. 5). Although ChIP-seq data concluded that GLK1 and GLK2 have the same binding profile[67], the transcriptional regulation of the two GLKs is different[68] and the *Arabidopsis thaliana* cell culture does not express *GLK1* during the greening process[19]. Possibly GLK1 and GLK2 play different roles where GLK2 is dominant regarding the regulation of chromatin.

Our data shows that the H3K27Ac-mediated activation mechanism at the *PhANG* loci needs a strong developmental cue, the GUN1-mediated retrograde signal (Figs. 3 and 4). The retrograde signalling protein GUN1 restricts chloroplast development in darkness and during early light response[19]. Following exposure to light GUN1 protein levels decrease and the suppression of a large number of transcription factors, including the GLKs, is released. However, whether there is a direct connection between VAL1, the Polycomb, H3K27me3 and GUN1 remains to be determined. The organelles are central in energy metabolism and our model clearly demonstrates that the epigenome is closely intertwined with organellar activities through retrograde signalling.

## Methods

### Plant material and transgenic lines

*Arabidopsis thaliana* accession Columbia-0 (Col-0) was used as wild-type. Seeds were obtained from the European Nottingham Arabidopsis Stock Centre (NASC) and have been previously described: *glk1;glk2* double mutant[38], *35S::GLK2-GFP* over expressor[55] and *val1-2*[69]. *glk1;glk2;val1* triple mutant was achieved by crossing and selection of homozygous lines by DNA genotyping. The *elf6-3* seeds and *clf-29* were kindly gifted by Dr. Jordi Moreno-Romero (UB) and Dr. Julia Qüesta (CRAG), respectively. *35S::VAL1:4xMYC (VAL1-ox)* was developed first by cloning the coding sequences from *VAL1* into pGWB517[70] via Gateway cloning, using pENTR™/D-TOPO™ and pDONR221 vectors, respectively (Thermo Fisher Scientific). Secondly, plant transformation was performed by floral dipping protocol using Agrobacterium tumefaciens (GV3101)-mediated transfer[71]. *35S::VAL1:4xMYC* was transformed into Col-0. Primers used for genotyping and cloning can be found in *Supplementary Data5*.

### Growth conditions

*Arabidopsis* cell culture was grown as in Dubreuil et al., 2018. Samples were collected by filtration. Treatments with 1 mM lincomycin were applied on Day0 with the fresh greening media before transferring culture to light. *Arabidopsis* plants were grown in a photoperiod of 16 hours light and 8 hours dark at 21 °C. *Arabidopsis* seeds were surface sterilized with ethanol and sodium hypochlorite and stratified at 4 °C for 3–5 days. Seedlings were grown on MS medium (0.5x Murashige and Skoog (MS) basal salt; 0.05 % MES; pH5.7) containing 0.6 % plant agar. Seedling (de)etiolation consisted of a first exposure of 4 hours of light after seed stratification, followed by 3 days of darkness growth at 21 °C. Dark-grown seedlings were collected in the dark under a green lamp or moved to a LED continuous light cabinet (100 μmol photons m − 2, s − 1) and harvested at specific time-points by liquid nitrogen snap freezing.

### Pigment quantification

Extraction was performed in two rounds to ensure complete extraction by adding 250 μl of cold methanol to 25 mg of grounded frozen sample, mixing, and leaving on ice in darkness for 5 minutes. Next, samples were centrifuged (5 min, 16,000 G, 4 °C). From the pooled 500ul, 400 μl was used to fill 2 wells (as technical replicates) in a 96-well microplate (83.3924, SARSTEDT). A470, A652, A665, and A750 absorbances were read with a SpectraMAX 190 microplate reader[72,73]. Quantification of pigment First, the absorbance was corrected for a volume of 200 μl MeOH in microplates. Chlorophyll A: Corrected A652c = (A652-A750)/0.58; Chlorophyll B: Corrected A665c= (A665-A750)/0.58; Carotenoids: Corrected A470c = (A470-A750)/0.58. Afterward, pigments were calculated: ChlA(μg/ml)=[−6.5079*A652c] + [16.2127*A665c]; ChlB(μg/ml)= [32.1228*A652c]−[13.8255*A665c]; Carotenoids(μg/ml) = [1000*A470c] − [1.63*ChlA] − [104.96*ChlB]/221. Finally, the total amount of pigments in μg/mg of fresh weight was recalculated. A750 was used as a blank.

### Greening phenotyping by scan

Scanning of plates containing seedlings at specific de-etiolation times was performed in a EPSON expression 12000XL scanner, using 24-bit RGB colour.TIFF format at a resolution of 2400 ppi, scaling output in 120 × 120 mm. Image scans were processed with ImageJ[74]. Before measurements, grey values were adjusted between 50 and 150 for each colour channel. RGB channels were split only to keep the 8-bit green channel. Region Of Interests (ROIs) were drawn on cotyledons, using only seedlings along the same plane. At least *n* = 25 seedlings per condition were used. To evaluate the green pixel value, the mean of the ROIs grey value was subtracted from the maximum grey value (255). GraphPad Prism was used for statistics and plotting. The mean green value was tested by One-way ANOVA followed by Tukey post-hoc test for multiple comparisons. Scale bar = 0.5 mm.

### mRNA quantification

Extraction and cDNA. Total RNA was extracted from frozen grounded powder from *Arabidopsis* culture or seedlings with the RNeasy Kit following the manufacturer's instructions (QIAGEN). Extracted RNA was treated with DNase I (Thermo Scientific™). Complementary DNA (cDNA) was synthesized using iScript™ cDNA Synthesis Kit (Bio-Rad) following the manufacturer's instructions. Quantitative qPCR reactions were performed using iQ™ SYBR® Green Supermix (Bio-Rad) on CFX96 or 384™ Real-Time System (Bio-Rad). Relative gene expression was calculated using the ΔCt method, normalized to the geometric mean of two control targets, and relative values were calculated for each control condition. Primers used for qPCR are listed in *Supplementary Data5*.

### Protein Gel Electrophoresis and Immunoblotting

Nuclei isolated as in[75], were resuspended with 4x Laemmeli sample buffer (278 mM Tris-HCl ph 6.8, 4.2% SDS, 44.4% Glycerol, 0.02% Bromophenol blue), heated 5 min at 90 °C, loaded to 10% acrylamide gel for SDS-PAGE separation and then transferred on a PVDF membrane using the Trans-Blot® Turbo™ Transfer System (Bio-Rad). Membrane blocking was performed for 1 h with 5% skim milk in TBST (20 mM Tris, 150 mM NaCl, 0,1% Tween) at RT. Blotting with antibodies against myc 1:1000 (Agrisera, AS15 3034) and incubating with anti-rabbit HRP-conjugated secondary antibody 1:10000 (Agrisera, AS09 602) or directly with anti-HA conjugated with peroxidase 1:2000 (Roche, 12013819001). Chemiluminescence was detected using the ECL kit (Agrisera, AS16 ECL-SN). For CBB staining, membranes were stained for 10 min using staining solution (0.1% Coomassie Brilliant Blue R250, 10% acetic acid and 50% methanol) and destained with distilled water.

### Chromatin immunoprecipitation

Chromatin preparation was achieved according to[75], with slight modifications. *Arabidopsis* single cell suspension culture was fixed at specific time points (dark and 1, 4 and 7 days in constant light) adding a final concentration of 0.75% formaldehyde into the shaking culture during 10 minutes and quenched with a final concentration of 125 mM Glycine during 10 minutes. Cells were recovered by filtration and washed twice with MC buffer. Two independent biological replicates for each time point were obtained by polling several flasks of cultured cells. Each biological replicate accounted for 2 grams of dry-weight cells which were first snap-frozen in liquid nitrogen. Arabidopsis seedling samples were first snap frozen in liquid nitrogen upon collection. After being ground into a fine powder, 500 mg of the sample was fixed in 20 ml of MC buffer with a final concentration of 0.75% formaldehyde for 10 min, quenched with a final concentration of 125 mM Glycine for 10 min and washed twice with MC buffer.

Extraction buffers were supplemented with 1% sucrose, M1 buffer was supplemented with 5 mM DTT and sonication buffer was supplemented with 2% SDS, while removing SDS from the IP buffer. Chromatin shearing was achieved with a focused-ultrasonicator (Covaris S2 system). DNA from a small aliquot of each chromatin extract was isolated, quantified and tested to contain 100–500-bp fragments. For immunoprecipitation, each biological replicate was divided into equal chromatin aliquots (100ug for ChIP-seq, 30 ug for ChIP-qPCR), while keeping a small fraction as Input. Chromatin was incubated overnight at 4 °C with anti-H3 (Agrisera, Cat. AS10710), anti-H3K4me3 (Millipore, Cat. 17-614), anti-H3K27me3 (Active Motif, Cat. 39155), anti-H3K27ac (Abcam, Cat. ab4729) or anti-H3K9me2 (Diagenode, Cat. C15410060) antibodies. The next day, prot-G magnetic dynabeads (Invitrogen, Cat. 10004D) were added for 4 h. After washes and elution, samples were incubated overnight at 55 °C adding proteinase K to a final concentration of 0.5 mg/ml. Purification of DNA after decrosslinking was performed by phenol:chloroform method.

For ChIP-qPCR, we used 3 samples (Input, H3 and the histone mark of interest) to quantify proper differences between conditions. At each target, the % of the signal relative to the Input of the histone mark of interest was divided by the % of the signal relative to the Input of H3. Then, the geometric mean of 2 control targets was used to further normalize the signal at photosynthesis targets. 2 sets of 2 control primers (Control1 and Control2 for H3K27ac plus Control3 and Control4 H3K27me3) were selected from regions with abundant and constant histone PTM signal through all time-points and conditions. Finally, the relative value at each target was calculated towards the control condition. Primers used for qPCR are listed in *Supplementary Data5*.

### ChIP-seq analysis

Mapping and peak calling. Sequence read preprocessing and quality assessment of the raw data were performed using FastQC-0.10.1 https://www.bioinformatics.babraham.ac.uk/projects/fastqc/for quality control. Reads were aligned to the *Arabidopsis* genome (TAIR10 release) using the software package BWA[76]. Trimmomatic was used for trimming and adapter removal[77]. The software MACS2[78] was used to call peaks using narrow-peak default parameters for H3K27ac and H3K4me3 and broad-peak default parameters for H3K27me3 and H3K9me2 marks for each replicate. Peak calling was performed using paired H3 and Input samples as controls, generating 2 types of peaks files, H3-control or Input-control. Only conserved peaks predicted using either Input and H3 as controls were kept for downstream analysis. Narrow peaks were extended ±250bp from the summit. Bed coordinates files were processed using bedtools[79]. Differential enrichment. We used all combined peak coordinates to evaluate each histone mark signal at each time-point, calculating differentially enriched regions (DERs) using the R package DiffBind using default parameters and H3 as control. Additionally, non-arbitrary profiles were designed allowing the analysis of dynamic behaviors through the experiment. Profile5 is characterized by non-significant changes in H3K27ac until the transition from Day4 to Day7 where the threshold was >2 fold increase and <0.05 FDR. Moreover, we integrated the 4 different histone marks data with our previously published RNA-seq data[3] by mapping the RNA-seq to the genome by STAR[80] and using bedtools coverage to extract RNA signal at each DER. DER/Gene annotation. DER bed files were intersected with Araport11 annotation. In case of multiple matches all loci where conserved. Transcription factor prediction. Sequences extracted within 1000 bp windows centered from each DER were used for DNA motif analysis. DNA motifs were inferred using HOMER Motif Analysis Algorithm[37] combining both de novo and *known* HOMER motifs. Then, DNA motifs were matched both to the HOMER library of reliable motifs and footprintDB[81]. A final list of transcription factors was curated considering their presence in the total sequences analyzed. Gene Ontology Enrichment and Tree maps. To determine whether any functional categories were overrepresented among genes among the several categories we identified, we performed a functional enrichment analysis of GO categories using GOFER (bschiffthaler/gofer3: Gofer3 v0.11. 2020). Go terms were summarized into Tree maps using REVIGO[82]. Visualization and heatmaps/metaprofiles. Bam coverage files were converted to bigwig[83] with a 50 bp read extension. 2 replicates were merged to visualize each condition. Tracks were visualized in IGV[84] exported as.png and formatted in Illustrator. Heatmaps and meta-profiles were constructed using bigwigs with deepTools[85].

**Published datasets** were used to support and validate our chromatin profiling results. H3K27ac and input control data were obtained from dataset GSE79524 (https://doi.org/10.1038/s41477-017-0023-7). H3K4me3 and input ChIP-seq data were sourced from GSE143831 (https://doi.org/10.1038/s41467-020-16651-5). H3K27me3, H3K9me2, HA control, and REF6 binding data were retrieved from dataset GSE65329 (https://doi.org/10.1038/ng.3556). ChIP-seq data for VAL1 were obtained from GSM4317615 (https://doi.org/10.1093/nar/gkaa1129). GLK2 ChIP-seq data were accessed from GSM6788601 (https://doi.org/10.1038/s41467-022-35438-4). Additional datasets profiling REF6 and H3K27me3 in Col-0 and *ref6* mutants were sourced from GSE181292 (https://doi.org/10.1093/nsr/nwab213), an independent REF6 dataset was accessed via GSM5106479 (https://doi.org/10.1016/j.jgg.2022.09.001), and RNA-seq from Arabidopsis cell culture via PRJEB21008 (https://doi.org/10.1104/pp.17.00435), which can be also found in Supplementary Data 1.

### Statistics & Reproducibility

No statistical method was used to predetermine sample size. All experiments were conducted using at least two or three independent biological replicates, as specified in figure legends and methods. No data were excluded from the analyses. The experiments were not randomized, and the investigators were not blinded to allocation during experiments and outcome assessment. For ChIP-qPCR, statistical significance between conditions was assessed using an unpaired two-sided Welch's t-test, and for multiple comparisons, Bonferroni correction was applied when appropriate. Greening phenotypes were evaluated using one-way ANOVA followed by Tukey's post hoc test. All statistical analyses were conducted using GraphPad Prism, excel or R.

### Reporting summary

Further information on research design is available in the Nature Portfolio Reporting Summary linked to this article.

### Data availability

Sequencing data generated in this study are available at at the European Nucleotide Archive (ENA) as accession PRJEB66263. Raw images for DNA and WB are deposited in Zenodo. https://doi.org/10.5281/zenodo.15678310. Plasmids and mutant lines generated in this study are available upon reasonable request to the corresponding authors. Source data are provided with this paper.

### Code availability

A detailed compilation of scripts and processing files can be found at: https://github.com/martiquevedo/Quevedo_et_al_2025. and are also deposited at https://doi.org/10.5281/zenodo.15674769.

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

## Acknowledgements

We are grateful to ABRC/NASC for *ref6-3* and *val1-2* seeds, to Jordi Moreno-Romero for *ref6-2, elf6-3* seeds and Julia Qüesta for *clf-29* seeds. We thank Xu Jin for technical assistance, the Umeå Plant Science Centre bioinformatics facility (https://www.upsc.se/platforms/upsc-bioinformaticsfacility.html) and Alonso R. Serrano for technical support with regards to the ChIP-sequencing data analysis. We thank DC SciArt (https://www.dariasciart.com/) for scientific illustration services. Part of this work was supported by grants and financial support to Å.S. from the Swedish Research Council, VR (2020-03958) and Foundation for Strategic Research, SSF (ARC19-0051) and to E.M. from FEDER/Ministerio de Ciencia, Innovación y Universidades – Agencia Estatal de Investigación (Project Reference PID2021-122288NB-I00); from the CERCA Programme/ Generalitat de Catalunya (Project Reference 2021SGR-792), and from the Spanish Ministry of Economy and Competitiveness, through the 'Severo Ochoa Programme for Centres of Excellence" in CEX2019-000902-S funded by MCIN/AEI/ 10.13039/501100011033. M.Q. received postdoctoral funding from the European Union's Horizon 2020 research and innovation programme under the Marie Skłodowska-Curie grant agreement no. 945043.

## Author contributions

M.Q. and Å.S. conceptualized the study and designed the experiments. M.Q., E.M. and Å.S. wrote the manuscript with significant input from all authors. M.Q., I.K., A.B. and L.C.C. performed the experimental work. All authors approved the final version of the manuscript.

## Funding

## Competing interests

The authors declare no competing interests.
