## [Transparent Peer Review file · Nature Communications]

Retrograde signals control dynamic changes to the chromatin state at photosynthesis-associated loci

Corresponding Author: Professor Åsa Strand

Version 0:

Reviewer comments:

Reviewer #1

(Remarks to the Author)

In this work, Quevedo and colleagues provide interesting insights into the histone modification patterns that occur during the transition of dark-grown *Arabidopsis* cell culture into a cell culture with functional chloroplast over the course of 7 days. The experiments all seem to have been performed to a high standard, and the presentation is in general very attractive. Comparison of the epigenetic changes match well with those of de-etiolated seedling, though at a much slower pace (1 day vs 7 days), suggesting the patterns are physiologically relevant. The authors then focus on the role of two classes of transcription factors, GLKs which are known to be required for chloroplast development, and VAL1 which rather is a repressor it seems. Overall the paper provides many interesting insights into epigenetic changes during greening.

My main concern is that I don't find the title claim that this is directly controlled by 'retrograde' signaling is supported. No evidence is provided that this is truly regulated by retrograde pathways (i.e. a signal emanating from a chloroplast that (in)directly steers gene expression, or in this case histone modifications). To me the observed modifications in histone PTMs can also be explained by indirect effects on metabolism, or other effects of having functional chloroplasts to 'reset' the epigenetic landscape. The 'retrograde' component refers to a previous study from the same group showing a two-phase response in the greening process of this culture (Dubreuil et al. 2018), but so much is happening in this phase that is hard to disentangle what is truly 'retrograde' and what is an effect of photosynthesis and chloroplast metabolism kicking in. On line 152 the authors state that after 24h of light the retrograde signal has been triggered. This is rather assumed, but no real data is provided for this. The experiments with lincomycin show that histone PTMs can indeed be blocked by blocking chloroplast ribosomes. I don't think this is however sufficient as evidence to claim that the histone modifications are in fact under retrograde control. This could be explained by any other pathway that would be affected by not having chloroplast development and ensuing photosynthesis. To show this, the authors would have to do light-induced histone PTM measurement experiments (e.g. Figure 3) in retrograde signalling mutants such as *gun1*, *gun5* or similar. If the histone patterns would be blocked, this would be better evidence for the claim that retrograde signals 'control' epigenetic changes.

Some further comments

- It is not very clear to me what Figure 1b is actually showing. The comparison in Supplementary figure 1d to me is much clearer to show the similarity of histone modifications between cell culture and seedlings. Please explain more clearly what the 'V' like diagrams are actually representing. Also the black text is very hard to read on the dark blue background of H3K27ac
- I also find the panel in Fig 4C very difficult to interpret, is each horizontal 'pixel' a different gene? How were the genes represented selected?
- In figure 4e a range of chlorophyll-related phenotypes are shown. Firstly it should be mentioned that the pale green phenotype in *glk 1 glk2* was already described previously (Waters et al., 2008, Plant J), so this not novel per se. The statistical analysis of Figure 4f is also not very clear. The authors describe a one-way ANOVA, but do not state which comparisons the * refer to (vs Col-0 in the same treatment?). Given that there is both a treatment (de-etiolation) and genotype (mutant vs WT) factor, a two-way ANOVA should be performed.
- The authors describe that the *glk1 glk2* mutants are T-DNA insertion mutants, while in fact they are dSpm transposon insertion mutants (Tissier et al., Plant Cell, 1999; Fitter et al., 2002, Plant J).
- The representation of histone PTMs vs VAL1 Chip-seq are quite clear to interpret from Fig 5a for the GLK1 locus (singular of loci is 'locus', not 'loci'). It would be good to show similar analysis for VAL1 and GLK2 Chip-seq data vs histone PTMs for

the LHCB2.2 and related marker genes in Figure 4g.

- The de-repressed expression of Phangs in the *glk1/2/val1* triple mutant is only observed for two genes. This data should also be shown for more classical retrograde markers like LHCB genes shown in e.g. Fig 4g.
- Line 229-230 the authors state that they have shown that Phangs are repressed at the chromatin level by maintaining repressive H3K27me3 state. This is rather an observed correlation, and the authors did not show that H3K27me3 is actually doing that in this case, which would require experiments in which the H3K27me3 is specifically removed.

Minor comments

- Some more information on the procedure for the measurement of histone PTMs (histone ChIP-seq) would be useful in the first Results paragraph.
- It is a bit confusing that the naming for GLKs and VAL1 in figure 4a and suppl Figure 5 are not matching, so it is not straightforward how these fit in with the other overrepresented sites
- Figure 2b, the fonts are very small and 'photosynthesis' blocks some of the underlying words.
- The expression levels should also be shown in Col-0 and GLK2OX in control conditions, not only under lincomycin conditions.

Reviewer #2

(Remarks to the Author)

The authors question the impact of chloroplast retrograde signaling at the chromatin level to regulate nuclear-encoded chloroplast genes (also called Photosynthesis Associated Nuclear Genes, PhANGs). They used dark-adapted Arabidopsis cell cultures in which the biogenesis of chloroplasts is triggered synchronously upon transfer to light. Through ChIP-seq of four histone marks (2 repressive and 2 activating) at 4 time points (0, 1, 4, and 7 days of continuous light), they identified that loss of H3K27me3 precedes gain of H3K27ac at PhANGs. After a motif prediction analysis and public ChIP-seq data mining, they propose that the transcription factors GLKs and VAL1 are linked to the dynamics of H3K27ac and H3K27me3, respectively. Their implication was confirmed on a subset of PhANGs in a line overexpressing VAL1 and in the double mutant *glk1;glk2*. De-etiolating seedlings are tested in parallel to validate the findings in planta and the authors propose the action of a retrograde signal using the plastid inhibitor lincomycin.

The impact of organelle retrograde signaling on the epigenome landscape has been barely studied, and as such the conclusions drawn in the present manuscript are of great interest to the readership of Nature Communications. However, while the experiments are well conducted and the results mostly well presented, they do not suffice to support conclusions about (1) a causal link between the dynamics at the chromatin level and the expression changes (only correlations between the two are presented but causality is not directly tested) and about (2) the involvement of a retrograde signal (lincomycin triggers retrograde signaling but also blocks photosynthesis and probably many other things, the observed effects might therefore be unrelated to retrograde signaling).

(1) On the causal links between the dynamics at the chromatin level and the expression changes:

- Line 18-19: "We asked whether such plastid retrograde signals control nuclear gene expression by altering the chromatin state during the establishment of photosynthetic function in response to light."
- Line 79-81: "This epigenetic reprogramming is essential for the activation of PhANGs during the second phase of the process of chloroplast development and for the establishment of photosynthetic activity".
- Lines 142-144: « Our data indicated that a group of photosynthesis genes is regulated by early repression and late acetylation as these PhANGs lost the repressive H3K27me3 mark on Day 4 and gained the H3K27ac mark on Day 7 »
In the absence of experiments on mutants affecting H3K27me3 and H3K27ac deposition/removal, it is not possible to infer causality between the effect on chromatin and the effect on expression.
 - For H3K27me3, *clf* and *elf6ref6* mutants (deposition and removal) must be tested for greening, PhANG expression, and H3K27ac gain in the cell culture system and in de-etiolating seedlings. The overexpressor of VAL1 is used but if H3K27me3 is elevated and retained on PhANGs after day 4 in culture cells or after 10h of light in de-etiolating seedlings in this line has not been checked.
 - For H3K27ac, *gcn5* could be a good candidate mutant to test for greening, PhANG expression, and H3K27me3 loss if H3K27ac at PhANGs is impaired in the mutant.

(2) On the involvement of a retrograde signal:

- Line 23, "is dependent on a plastid retrograde signal"
 - Line 77-79: "We discovered a mechanism by which retrograde signals trigger a specific switch in histone modification at photosynthesis associated loci."
 - Lines 167-168: "These observations support a mechanism by which retrograde signals modulate a switch in chromatin compaction at the photosynthesis loci."
 - Line 252: "We further demonstrated that the epigenome is closely connected to organellar activities by retrograde signals, and that a plastid signal is required for the specific changes to chromatin compaction"
 - Line 313: "Our data shows that the H3K27me3-mediated repressive mechanism at the PhANGs loci needs to be overcome by a strong developmental cue, the retrograde signal"
 - Line 319: "Our model clearly demonstrates that the epigenome is closely intertwined with organellar activities through retrograde signaling."
 - Line 321: "Retrograde signals trigger a specific switch in the histone code"
- The results that are presented show that lincomycin, an inhibitor of chloroplast translation, prevents the loss of H3K27me3

and the gain of H3K27ac between day 4 and day 7 in the cell culture or on 7-day-old light-grown seedlings in these conditions at 8 PhANGs by ChIP-qPCR. Lincomycin indeed triggers retrograde signaling to the nucleus but also has profound effects on the cells, so the involvement of such retrograde signaling in the dynamics of the chromatin marks is not demonstrated here. For example, lincomycin could prevent cell division and, as such, could impair H3K27me3 passive loss through replication (as suggested by the authors in line 262). As another example, H3K27me3 has been linked to TOR signaling, which is strongly reduced when photosynthesis is prevented as is the case with lincomycin. To demonstrate that a retrograde signal is involved, the authors need to show that the dynamics of H3K27me3 and H3K27ac are not affected by lincomycin in mutants of retrograde signaling, such as *gun1*.

Other comments:

The expressions "chromatin compaction", and "closed/open chromatin state" must be avoided as they are not assessed in the study.

The rationale to study group 5 genes vs the set of 105 genes (Figure 2d) vs the set of Day4-Day7 H3K27ac increased (figure 4c) is not clear and some discrepancy lies between the conclusions in the text and what is shown in the figures:

- On what criteria the genes were grouped in clusters in Supplemental Figure 2?
- How many of the 105 genes in 2d overlap with group 5 genes?
- Why use all the genes that increase H3K27ac in 4c and not the 105 genes that also lose H3K27me3?
- How many genes of these 3 sets are PhANGs? And oppositely, what is the proportion of all PhANGs undergoing loss of H3K27me3 followed by gain of H3K27ac?
- Line 140: "for this group of genes, the H3K27me3 mark was present in the Dark and Day1 but was lost before Day4", H3K27me3 on Group 5 genes is stable or increased in panel 2a and disagrees with this conclusion.

Figure 4e: The greening should be assessed also on cell cultures, in the set up that allowed their identification, and with a proper chlorophyll quantification rather than a RGB green measurements.

Line 239 : "overexpression of GLK2 was able to rescue the silenced chromatin state in conditions of dysfunctional chloroplasts and increase acetylation and transcription at the photosynthesis associated loci." - This conclusion is based on a first observation that lincomycin prevents loss of H3K27me3 (figure 3c and f). Why H3K27me3 was not assessed between day1 and day 4 in 3c is puzzling as this is the time window when the mark is lost, not between day 4 and day7 where loss has already happened. - Then it is based on a second observation that deposition of H3K27ac at PhANG loci under lincomycin is rescued in the GLK2ox line (Fig 6a). The untreated control is missing in the 6a panel to check how close the levels of H3K27ac reaches back to the levels of untreated plants.

Line 237-239: To sustain the assumption about a pioneer transcription factor function for GLK2, it must be checked first if the motifs bound by GLK2 in PhANG promoters are in a nucleosome-covered region before day 4 in culture cells, and in seedlings in the dark and under lincomycin treatment. This can be simply assessed from the H3 ChIP-seq data for culture cells, or ChIP-qPCR for the rest.

Line 303-304: "In addition to our results, reported greening phenotypes of lines over expressing the GLKs support pioneer properties of the GLKs". Non-pioneer TFs can explain the phenotype, in particular as overexpressing TF are notorious to bind to extra targets.

Line 304-305: "Our model suggests that a major demethylation event precedes GLK action" I don't think it is clear from the results if demethylation is required or not for GLK action as H3K27ac has been analyzed more systematically than H3K27me3. H3K27me3 marking should be quantified in VAL1ox, *glk1;2* in de-etiolating seedlings to check if a sustained H3K27me3 marking correlates with the decrease in H3K27ac observed (4g) and in *val1-2* and GLK2ox under lincomycin to check if a loss of H3K27me3 correlates with the rescued ability to gain H3K27ac (6a).

More generally, it must be better analyzed, in a systematic way, what is the overlap between VAL1 and GLK target genes, do they bind to the same promoters, to the same sites? Is it possible for GLKs to replace VAL1 at promoters?

Line 218-220: "Further support for the role of VAL1 controlling GLK1 expression was shown by the higher expression of GLK1 and PhANGs in the *glk1;glk2;val1* triple mutants compared to the *glk1;glk2* mutant following 10 hours of light exposure (Fig. 5d)."

I am confused about using the triple mutant *glk1;glk2;val1* to study the impact of VAL1 on GLK1 expression. The T-DNA insertion corresponding to the *glk1* mutation lies in the 5'UTR and so the VAL1 binding sites is apart from several kilobases from the GLK1 transcribed unit in this mutant. The effect of the *val1* mutation on GLK1 expression in the triple mutant *glk1;glk2;val1* is probably acting in trans rather than in cis in this genetic context.

Minor comments:

Where the 2 motifs chosen for Figure 4a originate from is unclear.

I do not see VAL1 binding in the supplementary figure 7d (Line 210-212).

line 260: Ref 49 is not about either H3K27ac or H3K27me3 and so I don't think hypotheses on methylation/acetylation rates can be inferred from this study.

Precise lincomycin condition in the title for Figure 6.

Peak annotation to genes is not described in the methods. It is not specified if motifs are searched in DERs or in the promoter of annotated genes from DERs.

Line 41, "a small number of TFs": light signaling on the contrary involves a plethora of transcription factors. Cell culture and seedlings cannot be equivalent (Line 103), even more so when studying PRC2 activity at genes that are tightly linked to cell differentiation. However, for the focus of the present study, H3K27me3 marking at all PhANGs could be systematically compared in the two systems.

Line 118: associated with growth in the absence of light not seen for H3K27ac and not the most significant term.

Line 121-123: Loss of H3K27me3 in cell wall synthesis genes, is it expected even in protoplasts

Why the GO terms in Supp1E are different from 1D for H3K27me3 hypomarked genes between Day1 and 4?

Line 150-151: I cannot see how the recovery supports a maintenance mechanism.

Figure 3 e and f: correct y scale label (compared to mock instead of day 4)

Figure 2a and Supplemental Figure 2: the normalization isn't precised in the legend or the methods. The color legend in Supplemental Figure 2 is missing.

Reviewer #3

(Remarks to the Author)

The manuscript by Quevedo et al., used an Arabidopsis cell culture to analyze the genome-wide dynamics of four histone marks during the acquisition of an operational photosynthesis. The authors found extensive epigenomic reprogramming throughout the chloroplast maturation period. Specifically, the transition from dark to light led to a reduction of H3K27me3 at over 2000 loci. A small portion of these genes (PhANGs) show a subsequent increase of the active mark H3K27ac. This switch requires an unknown retrograde signal, as well as the action of the sequence-specific PRC2 recruiter VAL1 and the positive-acting GLK TFs. Collectively, the authors found an interesting light-triggered transition of chromatin states regulated by chloroplast. However, the provided data is rather descriptive, thus more data is needed to better understand the observed phenomenon.

Major concerns

1. The overlap between genes with reduced H3K27me3 levels and genes with a H3K27ac increase is only 5%. The authors should discuss this finding in more detail. Are the 1900 genes with a H3K27ac direct GLK targets or why is no H3K27me3 repression required at these genes. The authors should indicate the degree of H3K27me3 depletion or H3K27ac gain within this groups and correlate it with gene expression changes. This might help to uncover which change is regulatory.
2. The discovered H3K27me3 depletion is striking but was not investigated. How is H3K27me3 removed. The authors should test the role of REF6 and potentially other H3K27me3 demethylases.
3. Whether GLKs must overcome a repressed or less accessible state of their CREs is unclear. It is more likely that the removal of H3K27me3 allows GLK binding. The authors should test H3K27me3 levels in their time course setting in glk double mutants and 35S:GLK plants.
4. It is unclear how the individual ChIP replicates were processed. Were they merged? It would be helpful to see the analyses of Fig.1 with the individual replicates. How do the replicates compare to each other?
5. In Figure 1b. It's unclear which genes and how many genes are shown in the individual heatmaps. How were they identified?
6. Figure 2a and 2g are hard to understand. H3K27me3 levels only go slightly down and then up again. Metagene plots are probably a better choice.

Reviewer #4

(Remarks to the Author)

Version 1:

Reviewer comments:

Reviewer #1

(Remarks to the Author)

The authors have added a lot of new and relevant data, with regards to e.g. the gun1 and ref6 mutants, and have overall

satisfactorily addressed my comments.

Some minor comments:

The new Figure 4 is nice. I think it would be easier to interpret if the actual relative H3K27ac and Phang Expression data were presented in control Col-0/gun1 and lincomycin Col-0/gun1, rather than just the normalized lincomycin only data.

Legends should be included close to the figures in the supplementary file, otherwise it gets harder to read.

Reviewer #2

(Remarks to the Author)

Please see the attached formatted pdf.

Reviewer #3

(Remarks to the Author)

The authors have made significant improvements to the manuscript, particularly with the inclusion of the new REF6 data. My previous concerns have been satisfactorily addressed. I now have only a few remaining concerns.

Major concerns:

1. The genome browser tracks in Supplementary Figure 7 should indicate whether VAL1 or REF6 peaks can be identified with MACS2 or comparable peak caller. The majority of VAL1 and REF6 peaks appear to be background rather than actual peaks.
2. A better option than REF6 ChIP-seq data could be H3K27me3 data in ref6 mutants since they display more than 1000 genomic regions that ectopically gain H3K27me3. Multiple datasets are already published and might help to further substantiate REF6-mediated regulation of Phang expression.

Minor concerns:

1. The gene lists in Supplementary Figure 6 need a better explanation. It's unclear which gene sets were used to create the metagene plots.
2. Do the authors have an explanation for the shift of VAL1 binding from upstream of the TSS to downstream of it in Figure 5C.

Version 2:

Reviewer comments:

Reviewer #2

(Remarks to the Author)

After two rounds of revision, the manuscript has been substantially improved. The authors provided the requested analysis of the gun1 mutant and followed the suggestion to replot ChIP-qPCR data using more appropriate control genes, which strengthened the findings on the effect of lincomycin, gun1 mutant, and REF6 overexpression, on gene regulation. Despite these improvements, the manuscript still displays a multitude of approximations and shortcomings in chromatin data analysis. Consequently, I have major concerns about important conclusions sustaining the proposed model that, in my opinion, is therefore not demonstrated.

The proposed model on a 2-step control of chromatin states at light-regulated genes is very appealing and would undoubtedly be of general interest to the readership of Nature Communications, yet the approximations and over-statements on key observations all together give the feeling that authors select the data that fit the model but are reluctant to test their accuracy using systematic analyses. This is seen in the lack of efforts to perform the requested consistent analyses of case study genes and also in the lack of genome-wide analyses empowering proper quantitative analyses of chromatin variations in the different samples.

Here is a summary of the significant issues raised but not addressed during the revision rounds:

- While claimed as being a major finding of the study (lines 136, 154, 163-164, 319, and the model in Figure 7), evidence for a first phase of H3K27me3 loss preceding a general gain in H3K27ac at RS-regulated genes during seedlings de-etiolation is still lacking. The results obtained with seedlings do not reproduce what is shown for the subset of 148 genes in a cultured cell system. In contrast to the claim, the data in Figure 2g and Figure Sup. 3 show either no change or a transient gain of H3K27me3 in response to light, hence if anything an increase but not a decrease. A demonstration would require, for example, a temporal and quantitative analysis of H3K27me3 and H3K27ac absolute levels and not just visual observations at a few genes.
- Although this was repeatedly requested, the study still lacks a systematic analysis of the overlap between the genes targeted by VAL1, REF6, and GLKs, and those losing/gaining H3K27me3 and H3K27ac. This hinders the evaluation of the authors' conclusions on the key concept of VAL1, REF6, and GLKs acting in a concerted manner at common specific loci. Instead, the manuscript displays heatmap signals, meta-profiles across a few selected genes and a few genome browser

views (Figure 5c, Sup. 6, Sup. 7, Sup. 8b) that, problematically, unconvincing support the claims of VAL1-REF6-GLK2 co-occurrence at chromatin. More precisely, most GLK2 and REF6 peaks are loosely defined or invisible. A demonstration would, for example, be based on correlation tests between the chromatin association of VAL1, GLK2, REF6 and the corresponding histone modifications for each of the VAL1 clusters analyzed, and/or determining how many genes are commonly bound by these proteins upon stringent peak calling (retrieved from the three studies) and the genes in the "overlap" subset.

- Some ChIP-qPCR data are still normalized to a control condition in which we ignore if the tested loci are enriched in the corresponding histone modification or protein. Addressing this point can give rise to drastically different interpretations. Here, in several instances, the control condition corresponds to an unmarked situation (i.e., H3K27ac in darkness in Figures 2a, 2g, Sup. 2, Supp 3 at day 4 and H3K27me3 at day 7 in Figure 3c). Therefore, the ChIP signal of a locus unmarked in the control condition is normalized to background noise and can only give aberrant results. For example, a ChIP signal that decreases as compared to a background signal in the control condition would only quantify noise differences.
- Similarly, in the absence of systematic analysis of H3K27me3 and H3K27ac levels in the starting (dark) and final (10h light) conditions, concluding on general effects in the mutant and overexpressor lines on H3K27ac enrichment by looking at a few selected genes fitting the model in Figure 5g is an over-simplification.
- In contrast to the claim on lane 161, the data do not give information on chromatin opening/closing: this has not been probed at all. Linking H3K27acetylation to chromatin openness is a shortcoming, the latter could be interestingly assessed using histone H3 ChIP-seq or, better, MNase-seq.
- The conclusion on reduced H3 occupancy at central positions of the 'H3K27ac/H3K27me3 overlap cluster' regions or VAL clusters 1-2 (lines 232-6 and Supplementary Figures 6c/d and 8d) is over-interpreted: without quantitative analyses made using the same genome coordinates in all figure panels and precise functional annotations of these regions, the observed lower H3 occupancy could correspond to the formation of nucleosome-free regions linked to increased transcription and H3K27ac gain away from VAL1/GLK2 binding. For example, H3K27ac and H3K27me3 enrichments notoriously occur at promoter/TSS and gene body domains, respectively.
- The elegant model builds on direct effects of VAL1 and REF6 on histone H3 acetylation at specific PhAN genes, yet these factors are known to contribute to regulating histone H3 methylation. No experiment supports such the claim of these factors mediating a H3K27me3-to-H3K27ac switch, especially if considering that H3K27me3 demethylation during seedling de-etiolation is not robustly supported by data and can hardly serve to conclude on a passive and rapid H3K27 acetylation provoked by H3 demethylation at PhANGs. The model should for example invoke and/or test the activity of a histone acetyltransferase.

Given the issues detailed above and their unsuccessful clarification upon three reviewing rounds, I can only advise the rejection of the manuscript despite its topic and proposed model being of great interest and originality.

Reviewer #3

(Remarks to the Author)

The authors satisfyingly revised their manuscript, and I have no further concerns.

In response to my first previous concern, I just want to emphasize that identifying peaks through visual inspection using a genome browser is inherently subjective and arbitrary. In contrast, applying thresholds that are widely accepted in the field and established by the ENCODE consortium provides an objective and standardized approach. Therefore, if a peak does not appear even under relaxed threshold settings, it should not be considered a real peak.

Reviewer #5

(Remarks to the Author)

Version 3:

Reviewer comments:

Reviewer #2

(Remarks to the Author)

Dear Editor,

The study presents relevant phenotypic defects of chromatin mutants and overexpressing lines related to greening, however, as detailed below, even upon 3 rounds of revisions, it falls short of convincingly reporting the proposed chromatin

mechanistic process.

The authors have in several instances acknowledged feedback and modified the text and the figures accordingly. They removed over-statements on key concepts about open/closed chromatin and on a proposed pioneer function of GLK transcription factors. They adjusted inadequate control regions for ChIP-qPCR, etc. These necessary corrections are appreciated, yet they raise concerns about how much the authors master the concepts and technical specificities of chromatin biology and epigenomics, which may fall away from their initial know-how.

On another note, the authors repeatedly deny the need to provide fast and simple data analysis and representations, which is also worrying. For instance, a systematic correlation analysis between the enrichment of chromatin factors and genes gaining/losing the H3K27ac and H3K27me3 marks could have been done by data mining from the corresponding publications. Also, qPCR, just normalized as the % of input signal, could also have been simply added as a supplementary figure. As pointed out by reviewer 3 and detailed below, most analyses are based on heatmaps or profile observations but lack proper comparisons of peak detection, which can be done using individual ChIP-seq data of different types and laboratories to determine which chromatin enrichments are statistically significant before comparing them.

Detailed feedback on critical concerns not addressed after three rounds of revision:

1) One of the study's main conclusions is that a first phase of general H3K27me3 loss precedes a general gain in H3K27ac at light-responsive genes, in particular at Photosynthesis-Associated-Nuclear-Genes (PhANGs). This claim is asserted on 148 genes in the cell culture system but is not assessed in the seedling system, which is the only natural system to consider. There is indeed less H3K27me3 and more H3K27ac at the photosynthetic genes analyzed by qPCR in the presence of lincomycin (Figure 3e & f), and a concomitant loss of H3K27me3/gain of H3K27me3 between 12 and 24h of light (Figure supp 3), but this does not indicate a causal effect. Given the number of time points analyzed, this also does not allow a temporal discrimination of events during plant greening.

2) The mechanistic model is based on the common targeting of genes by VAL1, REF6, and GLKs. As pointed out by reviewer 3, assessing chromatin association by visual inspection of ChIP profiles on genome browser or plotting them as heatmaps is useful but does not provide statistical support to the observations, hence remains subjective. As such, a systematic analysis of statistically relevant enriched regions for these 3 factors and for the H3K27ac and H3K27me3 histone modifications, employing commonly-used peak calling, is critically lacking. This simple analysis may allow the authors to propose a co-occurrence of these factors at target genes. Furthermore, a strict reviewer would request sequential ChIP analyses (Re-ChIP) to ensure proper co-occurrence of two proteins at the same chromatin loci in the same cells, but knowing the difficulty of this approach, a minimal correlation analysis was demanded, and even this was not provided.

3) The mechanistic model also relies on ChIP-qPCR analyses at a small subset of photosynthetic genes for which normalization is essential. The 3 rounds of normalization applied (versus H3 signal, versus signals at control genes, and versus a control condition) involve much manipulation of the data, hence potential biases. The variations seen on the graphs can not only come from variations in the studied marks but may just reflect variations in H3 density. Indeed, Figure supp. 6c shows that H3 occupancy fluctuates.

4) Finally, while phenotypic analyses indicate a role for H3K27 methylation during the greening process, the general model and the abstract of the manuscript focus on a general control of H3K27 acetylation. A potential link between H3K27 methylation and acetylation during the transition is very interesting, yet the working model should include the involvement of unknown HATs/HDACs. Despite previous requests, hypotheses on the matter are lacking since, to the best of my knowledge, H3K27me3 loss cannot trigger a gain in H3K27ac per se.

Reviewer #1 (Remarks to the Author)

1. I don't find the title claim that this is directly controlled by 'retrograde' signaling is supported.

The treatment with Lincomycin activates the retrograde signal and is commonly used to investigate the involvement of retrograde signals. However, you are quite right that a better confirmation would be to use one of the genomes uncoupled mutant. As GUN1 has been shown to play a critical role during the greening process and is also activated by the lincomycin treatment the appropriate mutant for this analysis is the *gun1* mutant. We have now performed experiments determining *PhANG* expression and H3K27ac/H3K27me3/H3 in the *gun1* mutant following treatment with lincomycin. The data is presented as the new Figure 4 and in Supplementary data Fig. S11f (H3K27me3). Following growth on lincomycin the *gun1* mutant showed increased H3K27Ac/H3 deposition compared to wild type. Interestingly, the response is similar to the response on lincomycin shown for the GLK2-ox line (Figure 7). The new data including the *gun1* mutant strongly support an involvement of a GUN1 mediated signal in the deposition of H3K27Ac during the greening process and we know believe our title is supported by experimental data.

2. It is not very clear to me what Figure 1b is actually showing. The comparison in Supplementary figure 1d to me is much clearer to show the similarity of histone modifications between cell culture and seedlings. Please explain more clearly what the 'V' like diagrams are actually representing. Also the black text is very hard to read on the dark blue background of H3K27ac

We apologize for not explaining the figure properly. We have rewritten the figure legend to more clearly explain how the figure should be interpreted. In addition, we have changed the colors used in the figure for clarity. We agree that Supplementary Fig. 1 is more visually attractive. However, Figure 1b gives a whole genomic view and also helps visualize overlap between different marks (for example, H3K27ac and H3K4me3 show similar spread). We sincerely hope our changes have made the interpretation of the figure straightforward.

3. I also find the panel in Fig 4C very difficult to interpret, is each horizontal 'pixel' a different gene? How were the genes represented selected?

We apologize for not explaining the figure properly. As our initial presentation of the different gene categories was found unclear, we have re-analyzed our ChIPseq data. In the revised manuscript we more clearly highlight the role of the overlap between regions losing H3K27me3 and gaining H3K27ac. To avoid confusion with the different overlaps we have created the new Supplementary Fig. S6 where we characterize 3 subsets of genes, 1) all regions gaining acetylation (Day4toDay7 acetylated), 2) profile05 (which only gain acetylation at Day7) and 3) the overlap (switch 27me3/27ac) via Metagene ChIP-seq profiles. Supplementary Fig. S6 also displays some examples of *PhANGs* from each group. We have rewritten the text to better explain the subsets.

Regarding heatmaps, we have rewritten the text and the figure legend to more clearly explain how the figure should be interpreted. You are correct though, that each line is a genomic region ("genes centered at the TSS"). Genes represented here are depicted in the Y axis (focusing on the overlap 27me3/27ac).

4. In figure 4e a range of chlorophyll-related phenotypes are shown. Firstly it should be mentioned that the pale green phenotype in *glk 1 glk2* was already described previously (Waters et al., 2008, Plant J), so this not novel per se. The statistical analysis of Figure 4f is also not very clear. The authors describe a one-way ANOVA, but do not state which comparisons the * refer to (vs Col-0 in the same treatment?). Given that there is both a treatment (de-etiolation) and genotype (mutant vs WT) factor, a two-way ANOVA should be performed.

We have cited the pale phenotype of *glk1glk2* and the double mutant is primarily used as a reference for the *glk1glk2val1*. We have clarified the statistics. We performed one-way ANOVA followed by Dunnett post-hoc comparison. We have large populations of seedlings, and the

variable follows normality. Tests have been performed at each time point separately (hence the One-way and not Two-way ANOVA). The comparisons are made with the corresponding Col0 as reference: Col0 0h for the 0h time point and Col0 10h for the 10h time point. This has been clarified in the Figure legend.

5. The authors describe that the *glk1 glk2* mutants are T-DNA insertion mutants, while in fact they are dSpm transposon insertion mutants (Tissier et al., Plant Cell, 1999; Fitter et al., 2002, Plant J).

The dSpm transposon insertion was displayed in the supplemental data but we have incorrectly referred to the line as a T-DNA insertion line in the text. We have corrected this mistake. Thank you.

6. The representation of histone PTMs vs VAL1 Chip-seq are quite clear to interpret from Fig 5a for the GLK1 locus (singular of loci is 'locus', not 'loci'). It would be good to show similar analysis for VAL1 and GLK2 Chip-seq data vs histone PTMs for the LHCB2.2 and related marker genes in Figure 4g.

This is a very good suggestion, and we have included this data in the revised version of the manuscript. We now provide 8 loci as a new Supplementary Fig. 7 for the *PhANGs* evaluated in our study with GLK2, VAL1 and REF6.

7. The de-repressed expression of *Phangs* in the *glk1/2/val1* mutant is only observed for two genes. This data should also be shown for more classical retrograde markers like LHCB genes shown in e.g. Fig 4g.

We have included data for *glk1glk2* and *glk1glk2val1* for more of the *PhANGs* in the revised Figure 6e. We also show the expression during dark conditions, which is the condition when VAL1 is active, and the new data show a strong effect of the *val1* mutation on *PhANG* expression the dark. The effect is less striking following 10 h light exposure and the data is shown in new Supplementary Figure S12.

8. Line 229-230 the authors state that they have shown that *Phangs* are repressed at the chromatin level by maintaining repressive H3K27me3 state. This is rather an observed correlation, and the authors did not show that H3K27me3 is actually doing that in this case, which would require experiments in which the H3K27me3 is specifically removed.

We have included analysis of mutant and transgenic lines for REF6, a Zinc finger with histone H3 lysine 27 demethylase activity. REF6 was also identified using the HOMER Motif Analysis Algorithm of the temporal DERs or profile clusters. The *ref6* mutant and a REF6-ox line have now been included to all the panels of new Figure 5 and supplementary Figures 8, 9, 10, and 11. It is clear that REF6 plays a role in the transition from H3K27me3 to H3K27ac. The *ref6* mutant shows a pale phenotype, accumulates less chlorophyll, attenuated *PhANG* expression and reduced H3K27Ac deposition compared to wild type. We also included REF6-ox in the lincomycin experiment (New Supplementary Figure S11) and the lines display similar phenotype to the GLK2-ox line.

9. Some more information on the procedure for the measurement of histone PTMs (histone Chip-seq) would be useful in the first Results paragraph.

We have changed the text to include a better description of the measurements of the histone PTMs. We have included the following lines at the second paragraph in the methods section: "We curated a consensus peak set for all histone marks and time-points and calculated differential binding affinity between the PTM CHIP-seq data and their paired H3 control. This allowed normalization for nucleosome occupancy by comparing PTM signals with total H3 levels. Our differential analysis revealed three distinct phases at the chromatin level".

10. It is a bit confusing that the naming for GLKs and VAL1 in figure 4a and suppl Figure 5 are not matching, so it is not straightforward how these fit in with the other overrepresented sites
We apologize and the mistakes have been corrected.

11. Figure 2b, the fonts are very small and 'photosynthesis' blocks some of the underlying words.
We have corrected this and changed the font. We have also redone the analysis for the overlap and the GO tree map looks slightly different.

12. The expression levels should also be shown in Col-0 and GLK2OX in control conditions, not only under lincomycin conditions
Data for GLK2-ox in control conditions is now shown in new Supplementary Figure S12b. The GLK2-ox display slightly elevated *PhANG* expression, but the effect is not comparable to the effect seen following growth on lincomycin.

Reviewer #2 (Remarks to the Author):

The impact of organelle retrograde signaling on the epigenome landscape has been barely studied, and as such the conclusions drawn in the present manuscript are of great interest to the readership of Nature Communications. However, while the experiments are well conducted and the results mostly well presented, they do not suffice to support conclusions about (1) a causal link between the dynamics at the chromatin level and the expression changes (only correlations between the two are presented but causality is not directly tested) and about (2) the involvement of a retrograde signal (lincomycin triggers retrograde signaling but also blocks photosynthesis and probably many other things, the observed effects might therefore be unrelated to retrograde signaling).

Major comments:

- (1) On the causal links between the dynamics at the chromatin level and the expression changes:
- Line 18-19: "We asked whether such plastid retrograde signals control nuclear gene expression by altering the chromatin state during the establishment of photosynthetic function in response to light."
 - Line 79-81: "This epigenetic reprogramming is essential for the activation of *PhANGs* during the second phase of the process of chloroplast development and for the establishment of photosynthetic activity".
 - Lines 142-144: « Our data indicated that a group of photosynthesis genes is regulated by early repression and late acetylation as these *PhANGs* lost the repressive H3K27me3 mark on Day 4 and gained the H3K27ac mark on Day 7 »
- In the absence of experiments on mutants affecting H3K27me3 and H3K27ac deposition/removal, it is not possible to infer causality between the effect on chromatin and the effect on expression. - For H3K27me3, *clf* and *elf6ref6* mutants (deposition and removal) must be tested for greening, *PhANG* expression, and H3K27ac gain in the cell culture system and in de-etiolating seedlings. The overexpressor of VAL1 is used but if H3K27me3 is elevated and retained on *PhANGs* after day 4 in culture cells or after 10h of light in de-etiolating seedlings in this line has not been checked.

To address the absence of experiments on mutants affecting H3K27me3 removal we have collected the *ref6*, *clf* and *elf6* mutants, all well characterized mutants shown to be affected in the H3K27me3 levels. The mutants were characterized during the greening process. This data is displayed in the new Supplementary Figure S11. Curiously, only the *ref6-2* and *ref6-3* demonstrated a significantly impaired greening process as shown by chlorophyll accumulation and *PhANG* expression (Supplementary Figure S11). Thus, we focused our further analysis on the REF6, a Zinc finger with histone H3 lysine 27 demethylase activity. REF6 was also identified using the HOMER Motif Analysis Algorithm of the temporal DERs or profile clusters (now shown in revised Figure 5 and Supplementary Figure S5). REF6 and the *ref6* mutant have now been included to all the panels of new Figure 5. It is clear that REF6 plays a role in the transition from H3K27me3 to H3K27ac. The *ref6* mutant shows a pale phenotype, accumulates less chlorophyll, attenuated *PhANG* expression and reduced H3K27Ac deposition

compared to wild type. We also included REF6-ox in the lincomycin experiment (New Supplementary Figure S11) and REF6-ox displayed similar phenotype to the GLK2-ox line.

- (2) On the involvement of a retrograde signal: • Line23, “is dependent on a plastid retrograde signal”
- Line 77-79: “We discovered a mechanism by which retrograde signals trigger a specific switch in histone modification at photosynthesis associated loci.”
 - Lines 167-168: “These observations support a mechanism by which retrograde signals modulate a switch in chromatin compaction at the photosynthesis loci.”
 - Line 252: “We further demonstrated that the epigenome is closely connected to organellar activities by retrograde signals, and that a plastid signal is required for the specific changes to chromatin compaction”
 - Line 313: “Our data shows that the H3K27me3-mediated repressive mechanism at the PhANGs loci needs to be overcome by a strong developmental cue, the retrograde signal”
 - Line 319: “Our model clearly demonstrates that the epigenome is closely intertwined with organellar activities through retrograde signaling.”
 - Line 321: “Retrograde signals trigger a specific switch in the histone code”. The results that are presented show that lincomycin, an inhibitor of chloroplast translation, prevents the loss of H3K27me3 and the gain of H3K2ac between day 4 and day 7 in the cell culture or on 7-day-old light-grown seedlings in these conditions at 8 PhANGs by ChIP-qPCR. Lincomycin indeed triggers retrograde signaling to the nucleus but also has profound effects on the cells, so the involvement of such retrograde signaling in the dynamics of the chromatin marks is not demonstrated here. For example, lincomycin could prevent cell division and, as such, could impair H3K27me3 passive loss through replication (as suggested by the authors in line 262). As another example, H3K27me3 has been linked to TOR signaling, which is strongly reduced when photosynthesis is prevented as is the case with lincomycin. To demonstrate that a retrograde signal is involved, the authors need to show that the dynamics of H3K27me3 and H3K27ac are not affected by lincomycin in mutants of retrograde signaling, such as *gun1*.

The treatment with Lincomycin activates the retrograde signal and is commonly used to investigate the involvement of retrograde signals. However, you are quite right that a better confirmation would be to use one of the genomes uncoupled mutant. As GUN1 has been shown to play a critical role during the greening process and is also activated by the lincomycin treatment the appropriate mutant for this analysis is the *gun1* mutant. We have now performed experiments determining *PhANG* expression and H3K27ac/H3K27me3/H3 in the *gun1* mutant following treatment with lincomycin. The new data is presented as the new Figure 4 and in Supplementary data Fig. S11 (H3K27me3). Following growth on lincomycin the *gun1* mutant show increased H3K27Ac/H3 deposition compared to wild type. Interestingly, the response is similar to the response on lincomycin shown for the GLK2-ox line (Figure 7). The new data including the *gun1* mutant strongly support an involvement of a GUN1 mediated signal in the deposition of H3K27Ac during the greening process.

Other comments:

3. The expressions “chromatin compaction”, and “closed/open chromatin state” must be avoided as they are not assessed in the study.

You are correct, this is an overstatement, and we have changed our wording.

4. The rationale to study group 5 genes vs the set of 105 genes (Figure 2d) vs the set of Day4-Day7 H3K27ac increased (figure 4c) is not clear and some discrepancy lies between the conclusions in the text and what is shown in the figures: - On what criteria the genes were grouped in clusters in Supplemental Figure 2?

We have re-analyzed the subsets and found that the “overlap” subset was bigger than what previously displayed, which now is corrected in the revised version of Figure 2. We realize we were not clear about the different gene groups, and we apologize for this. To avoid confusion

with the different subsets we have created the new Supplementary Figure 6 where we characterize 3 subsets of genes via metagene ChIP-seq profiles. To clarify we focus on: 1) all regions gaining acetylation (Day4 to Day7 acetylated), 2) profile05 (which only gain acetylation at Day7) and 3) the overlap (the switch 27me3/27ac). We have rewritten the text to better explain the subsets. The clusters shown in Supplementary Fig. S2 were non-arbitrary combinations of H3K27ac trajectories from the complete whole genome ChIPseq data set. The analysis was performed to check if in our data set the H3K27ac trajectories match RNA expression profiles and in addition, to identify profiles of interest, such as “acetylation” only after at Day4 (profile5). Each profile applies thresholds of at least 2-fold change at the time-points of interest.

5. How many of the 105 genes in 2d overlap with group 5 genes?

There are 23 loci that are shared between “overlap” and group 5. The new Supplementary Figure 6 shows some representative *PhANGs* present in each group. The complete gene lists can be found in the Supplementary tables

6. Why use all the genes that increase H3K27ac in 4c and not the 105 genes that also lose H3K27me3?

We agree and we were not clear in the previous versions of the manuscript about the three different categories. In the revised version of the manuscript, we characterized each subset in detail with special emphasis on the “overlap”. We have changed Figure 5 to only show the heatmaps of the “overlap” which is, as pointed out, the most interesting subset.

7. How many genes of these 3 sets are *PhANGs*? And oppositely, what is the proportion of all *PhANGs* undergoing loss of H3K27me3 followed by gain of H3K27ac?

All 3 sets are enriched by GO terms of photosynthesis. The definition of *PhANG* is however, somewhat vague, but a significant number (30 genes) of nuclear-encoded proteins directly associated with the photosynthetic reactions undergo the loss of H3K27me3 followed by gain of H3K27ac. In the new Supplementary Fig. S6 the 3 gene groups are shown and some examples of *PhANGs* from each group are displayed.

8. Line 140: “for this group of genes, the H3K27me3 mark was present in the Dark and Day1 but was lost before Day4”, H3K27me3 on Group 5 genes is stable or increased in panel 2a and disagrees with this conclusion.

We have corrected this mistake in the text and re-written the text with emphasis on the “overlap” group. We also present Metaplots in the new Supplementary Fig S6 that more clearly display the balance between H3K27ac/H3K27me3.

9. Figure 4e: The greening should be assessed also on cell cultures, in the set up that allowed their identification, and with a proper chlorophyll quantification rather than a RGB green measurements.

The greening process of the cell culture has been described in great detail previously, e.g. Dubreuil et al., *Plant Physiol.* (2018), Diaz et al., *Nature Comm.* (2018) and Hernandez et al., *New Phytol.* (2022).

10. Line 239 : “overexpression of GLK2 was able to rescue the silenced chromatin state in conditions of dysfunctional chloroplasts and increase acetylation and transcription at the photosynthesis associated loci.” - This conclusion is based on a first observation that lincomycin prevents loss of H3K27me3 (figure 3c and f). Why H3K27me3 was not assessed between day1 and day 4 in 3c is puzzling as this is the time window when the mark is lost, not between day 4 and day7 where loss has already happened.

In the cell culture we determined H3K27me3 and H3K27Ac at day 4 and day 7. The acetylation takes place after Day 4 in the cell culture and at day 7, H3K27me3 levels are low and H3K27Ac high. The point of this experiment was to test the involvement of a retrograde signal in this process and the retrograde signal is released sometime between day4 and day5 (Dubreuil et

al., *Plant Physiol.* (2018)). We chose the time points to stay clear of the key events to be able to detect potential differences in the PTMs. When we examined the PTMs in the *gun1* mutant we found that the major difference compared to wild type was the in deposition of H3K27Ac (shown in new figure 4). However no significant difference was found compared to wild type for the H3K27me3 levels (shown in new Supplementary Figure S11). We can only speculate that the recovery of H3K27me3 levels in the *gun1* mutant during growth on lincomycin could be explained by a described maintenance mechanism for a closed chromatin state as suggested by Mosquna et al., *Development*, (2009). We have also included a section in the discussion about this.

11. Then it is based on a second observation that deposition of H3K27ac at PhANG loci under lincomycin is rescued in the GLK2ox line (Fig 6a). The untreated control is missing in the 6a panel to check how close the levels of H3K27ac reaches back to the levels of untreated plants.

We provide data for the *PhANG* expression for GLK2-ox grown under control conditions (new Supplementary Figure S12). The GLK2-ox display slightly elevated *PhANG* expression compared to wildtype under control conditions, but the effect is not comparable to the effect seen following growth on lincomycin. If we compared the expression levels between control conditions and lincomycin conditions the expression is about 10% of the control levels in the *gun1* mutant and the GLK2-ox. In the wild type of course much less. Thus, most likely the levels of H3K27ac will be quite far off the control levels.

12.Line 237-239: To sustain the assumption about a pioneer transcription factor function for GLK2, it must be checked first if the motifs bound by GLK2 in PhANG promoters are in a nucleosome-covered region before day 4 in culture cells, and in seedlings in the dark and under lincomycin treatment. This can be simply assessed from the H3 ChIP-seq data for culture cells, or ChIP-qPCR for the rest.

This is a very good suggestion, thank you. We have performed panH3 analysis from our ChIP-seq data. The analyses show high H3 density during dark and following 1day light exposure. Those regions correlate with regions which are also bound by VAL. The H3 density is reduced at GLK targets from day4 and day7 and shows more accessible (less nucleosome dense) chromatin. The new data is shown in Supplementary Figures S6 and S8.

13.Line 303-304: "In addition to our results, reported greening phenotypes of lines over expressing the GLKs support pioneer properties of the GLKs". Non-pioneer TFs can explain the phenotype, in particular as overexpressing TF are notorious to bind to extra targets

You are correct, this was an overstatement. We have removed this statement.

14.Line 304-305: "Our model suggests that a major demethylation event precedes GLK action" I don't think it is clear from the results if demethylation is required or not for GLK action as H3K27ac has been analyzed more systematically than H3K27me3. H3K27me3 marking should be quantified in VAL1ox, glk1;2 in de-etiolating seedlings to check if a sustained H3K27me3 marking correlates with the decrease in H3K27ac observed (4g) and in val1-2 and GLK2ox under lincomycin to check if a loss of H3K27me3 correlates with the rescued ability to gain H3K27ac (6a).

We have tried but H3K27me3 was extremely difficult to monitor during deetiolation as its deposition and removal appear to be slower compared to H3K27ac. As described above, for the *gun1* mutant no significant difference was found compared to wild type for the H3K27me3 levels (shown in new Supplementary Figure S11). We can only speculate that the on-and-off behaviour of H3K27me3 deposition which will result in a mix K27me3/K27ac PTMs, could be explained by a maintenance mechanism for a closed chromatin state as suggested by Mosquna et al., *Development*, (2009). Furthermore, it has been reported that slow chromatin dynamics mediated by Polycomb act as a noise cancellation mechanism to avoid spontaneous activation of developmental programs (Berry et al., 2017) which is in line with the developmental checkpoint associated with the retrograde signal allowing coordination of the activities of the nuclear and plastid genomes. Single cell analysis might provide an answer to this question.

15. More generally, it must be better analyzed, in a systematic way, what is the overlap between VAL1 and GLK target genes, do they bind to the same promoters, to the same sites? Is it possible for GLKs to replace VAL1 at promoters

In the new Supplementary Figure S7 we now provide 8 *PhANG* loci matched with interaction data for GLK2, VAL1 and REF6. We have also profiled the binding of GLK2, VAL1 and REF6 at the 3 different subsets in Supplementary Figure S6. We can clearly assess that while GLK2 and VAL1 are found around the same genes, their peaks do not co-localize suggesting GLKs do not replace VAL1.

16. Line 218-220: "Further support for the role of VAL1 controlling GLK1 expression was shown by the higher expression of GLK1 and PhANGs in the *glk1;glk2;val1* triple mutants compared to the *glk1;glk2* mutant following 10 hours of light exposure (Fig. 5d)." I am confused about using the triple mutant *glk1;glk2;val1* to study the impact of VAL1 on GLK1 expression. The T-DNA insertion corresponding to the *glk1* mutation lies in the 5'UTR and so the VAL1 binding sites is apart from several kilobases from the GLK1 transcribed unit in this mutant. The effect of the *val1* mutation on GLK1 expression in the triple mutant *glk1;glk2;val1* is probably acting in trans rather than in cis in this genetic context.

We hypothesize that VAL1 binding downstream of *GLK1* affects its repression. How this VAL1-repression loop acts will be explored in the future. Nevertheless, we show clearly that VAL1-ox has a direct effect on *GLK1* expression. We further show in Figure 6a that the VAL1 peak is downstream 3'UTR. We have highlighted the direction of expression in the figure which may have caused the confusion. We apologize for this error.

Minor comments:

Where the 2 motifs chosen for Figure 4a originate from is unclear.

The mistakes have been corrected. We have included the names of GLKs, VAL and REF6 in Supplementary Figure S5.

I do not see VAL1 binding in the supplementary figure 7d (Line 210-212).

We have adjusted the background and provide several other *PhANGs* in Supplementary Figure S7

line 260: Ref 49 is not about either H3K27ac or H3K27me3 and so I don't think hypotheses on methylation/acetylation rates can be inferred from this study.

We have corrected the reference.

Precise lincomycin condition in the title for Figure 6.

We have re-written the title.

Peak annotation to genes is not described in the methods. It is not specified if motifs are searched in DERs or in the promoter of annotated genes from DERs.

We apologize for the mistake. Gene annotation was performed by intersecting DERs with *Araport11*, in case multiple loci fall in a DER all were kept. Motifs are searched in DERs, specifically within a window of 1000bp centered at each DER center. We have provided more details in the methods section.

Line 41, "a small number of TFs": light signaling on the contrary involves a plethora of transcription factors.

We have rewritten as "specialized set of TFs".

Cell culture and seedlings cannot be equivalent (Line 103), even more so when studying PRC2 activity at genes that are tightly linked to cell differentiation. However, for the focus of the present study, H3K27me3 marking at all *PhANGs* could be systematically compared in the two systems.

We agree, we used the word equivalent in terms of the histone PTM occupancy where we observed a significant overlap between seedlings and the cell culture at day7 (once the culture has differentiated into photosynthetically active cells). However, we have re-phrased the text accordingly.

Line 118: associated with growth in the absence of light not seen for H3K27ac and not the most significant term.

We have changed to growth in darkness.

Line 121-123: Loss of H3K27me3 in cell wall synthesis genes, is it expected even in protoplasts. **The cell line is not a protoplast it is in fact a cell line with a cell wall (Dubreuil et al., 2018).**

Why the GO terms in Supp1E are different from 1D for H3K27me3 hypomarked genes between Day1 and 4?

Supplementary Figure 1e is molecular function, 1d are only Biological process.

Line 150-151: I cannot see how the recovery supports a maintenance mechanism.

The on-and-off behaviour of H3K27me3 deposition could be explained by a maintenance mechanism for a closed chromatin state as suggested by Mosquna et al., Development, (2009).

Figure 3 e and f: correct y scale label (compared to mock instead of day 4)

Thanks, we have corrected the figure.

Figure 2a and Supplemental Figure 2: the normalization isn't precised in the legend or the methods. The color legend in Supplemental Figure 2 is missing.

The data is normalized to the Dark time-point, we have changed the legend.

Reviewer #3 (Remarks to the Author):

Major concerns.

1. The overlap between genes with reduced H3K27me3 levels and genes with a H3K27ac increase is only 5%. The authors should discuss this finding in more detail. Are the 1900 genes with a H3K27ac direct GLK targets or why is no H3K27me3 repression required at these genes. The authors should indicate the degree of H3K27me3 depletion or H3K27ac gain within this groups and correlate it with gene expression changes. This might help to uncover which change is regulatory.

Thank you for the comment. We have re-analyzed our data and found that the overlap was not calculated properly (we used DERs and not annotated loci). The subsets have been corrected. With this reanalysis, we have found the overlap to be of more importance and have provided several additional Figures to the manuscript.

To avoid confusion with the different overlaps we have created the new Supplementary Figure S6 where we characterize the 3 subsets of genes via metagene ChIP-seq profiles. To clarify we focus on: 1) all regions gaining acetylation (Day4toDay7 acetylated), 2) profile05 (which only gain acetylation at Day7) and 3) the overlap (switch 27me3/27ac). We have rewritten the text to better explain these subsets. We further characterize each subset looking at the profile of histone PTMs across time, panH3 and VAL1, GLK2 and REF6 binding.

2. The discovered H3K27me3 depletion is striking but was not investigated. How is H3K27me3 removed. The authors should test the role of REF6 and potentially other H3K27me3 demethylases.

To address the absence of experiments on mutants affecting H3K27me3 removal we have collected the *ref6*, *clf* and *elf6* mutants, all well characterized mutants shown to be affected in the H3K27me3 levels. The mutants were characterized during the greening process. This data is displayed in the new Supplementary Figure S11. Curiously, only the *ref6-2* and *ref6-3*

demonstrated significantly impaired greening process as shown by chlorophyll accumulation and PhANG expression (Supplementary Figure S11). Thus, we focused our further analysis on the REF6, a Zinc finger with histone H3 lysine 27 demethylase activity. REF6 was also identified using the HOMER Motif Analysis Algorithm of the temporal DERs or profile clusters (now shown in revised Figure 5 and Supplementary Figure S5). REF6 and the *ref6* mutant have now been included to all the panels of new Figure 5. It is clear that REF6 plays a role in the transition from H3K27me3 to H3K27ac. The *ref6* mutant shows a pale phenotype, accumulates less chlorophyll, attenuated *PhANG* expression and reduced H3K27Ac deposition compared to wild type. We also included REF6-ox in the lincomycin experiment (New Supplementary Figure S11) and REF6-ox displayed similar phenotype to the GLK2-ox line.

3. Whether GLKs must overcome a repressed or less accessible state of their CREs is unclear. It is more likely that the removal of H3K27me3 allows GLK binding. The authors should test H3K27me3 levels in their time course setting in *glk* double mutants and 35S:GLK plants.

We have tried but H3K27me3 was extremely difficult to reliably monitor during de-etiolation as its deposition and removal appear to be slower compared to H3K27ac. For the *gun1* mutant no significant difference was found compared to wild type for the H3K27me3 levels (shown in new Supplementary Figure S11). We can only speculate that the on-and-off behaviour of H3K27me3 deposition could be explained by a maintenance mechanism for a closed chromatin state as suggested by Mosquna et al., Development, (2009). Furthermore, it has been reported that slow chromatin dynamics mediated by Polycomb act as a noise cancellation mechanism to avoid spontaneous activation of developmental programs (Berry et al., 2017) which is in line with the developmental checkpoint associated with the retrograde signal allowing coordination of the activities of the nuclear and plastid genomes. We have included a section in the discussion about this.

4. It is unclear how the individual ChIP replicates were processed. Were they merged? It would be helpful to see the analyses of Fig.1 with the individual replicates. How do the replicates compare to each other?

We apologize for being unclear and have now provided detailed information in the methods section. We have exchanged Supplementary Figure 1a so that it now displays the behavior of the replicates (which are highly consistent).

5. In Figure 1b. It's unclear which genes and how many genes are shown in the individual heatmaps. How were they identified?

We display all DERs identified for each histone mark (only Day7). We have re-written the legend to make this clear.

6. Figure 2a and 2g are hard to understand. H3K27me3 levels only go slightly down and then up again. Metagene plots are probably a better choice.

We now present the data in Figure 2 in metaplots, Supplementary Figures S6 and S8. The data presented in Figure 2g is not RNAseq but qPCR data, we have clarified this in the legend.

Response to reviewers comments:**Reviewer 1**

The authors have added a lot of new and relevant data, with regards to e.g. the *gun1* and *ref6* mutants, and have overall satisfactorily addressed my comments.

-Thank you for your positive comments!

The new Figure 4 is nice. I think it would be easier to interpret if the actual relative H3K27ac and Phang Expression data were presented in control Col-0/*gun1* and lincomycin Col-0/*gun1*, rather than just the normalized lincomycin only data.

-The ChIP experiment was run on lincomycin treated material only. Thus, the comparison between the genotypes is the appropriate comparisons. However, in the Supplementary Figures S4 and S12 we show the expression levels for *PhANGs* in control conditions vs Lincomycin.

Legends should be included close to the figures in the supplementary file, otherwise it gets harder to read.

-Thank you for the comment. In the revised version of the Supplementary Figures, we have included the legends associated to the Figures.

Reviewer #2

After the improvement of the manuscript and a better description of the ChIP-qPCR procedure, I now have serious concerns about the ChIP-qPCR data normalization (detailed below), which, presented as it is, calls into question the validity of essential claims. About ChIP-qPCR data normalization. The ChIP-qPCR graphs (Figure 2 g, Figure 3 e, and f, Figure 4a, Figure 5g, Figure 7a, Supplemental Figures 3, 11e, 11f) result from a series of 4 normalization rounds that can introduce several biases: 1. The ChIP-qPCR values are calculated first, as usually done, as a the percentage of input². These values are then normalized against the % of input of an H3 IP on the same genomic target³. This 2-step normalization is then also applied to two control genes, UBC and EF-1alpha. The geometric mean of the values for these two genes serves then to normalize the previous value (obtained in step 2). This leads to a first caveat: as shown by a survey of the ChIP tracks provided with the study for the culture system, UBC and EF-1alpha are not marked by H3K27me₃ but only by H3K27ac in their promoter. Both primer pairs amplify promoter regions that are unfortunately devoid of H3K27me₃. Therefore, I am afraid that all H3K27me₃ qPCR data presented in the article is normalized to background noise.

Finally, a fourth round of normalization is made towards the value obtained in the control condition. These control conditions are, depending on the figure, the WT dark (2g and supp3), WT light mock (3ef,5g), or WT light Lincomycin (4a,7a, supp11ef). Here comes the second caveat: The 8 genes assessed have been selected for differential marking in the cell culture system, yet proper H3K27ac and H3K27me₃ marking have not been verified for each of the three control conditions. If a gene is not marked in the control condition (denominator), then the ratio is a ratio over background noise. This essential point should be checked. Overall, this 4-step normalization procedure assumes twice that the level of background signal is reproducible between different genome positions, samples, and replicates. As such, there is a reasonable doubt that the observed differences report actual differences in marking. To avoid biases, qPCR data are usually presented simply as % of input (as such, or relative to the percentage of input of the H3 signal). Moreover, for each histone modification, qPCR figures should display side by side the different time points, culture conditions, or genotypes. Importantly, each qPCR analysis should also include at least one unmarked locus to compare proper signals from background levels, and a marked locus to compare the validity of peak analysis.

- Thank you for this very insightful comment on our qPCR analysis. While we agree that ChIP reports can be represented as raw % of input, we find that normalization by H3 and background regions correct for differences in ChIP efficiencies between different conditions and samples. In addition, this normalization method better detect subtle changes in histone PTMs. In addition, we want to emphasize that our ChIP pipeline originates from an initial step of accurately correcting the concentration of chromatin used in each ChIP reaction to minimize technical variations such as usage of antibody/beads/input in every experiment.

In response to your concerns, we have addressed the point made about our original control primers for H3K27me3 targeting background and as you suggested we have re-run our ChIP-qPCR using new control primers targeting enriched H3K27me3 stable regions.

The primers have been selected to have constant enrichment for H3K27me3 during the greening process of the culture, and in Arabidopsis seedlings while being depleted of H3K27ac. These primers have been named control3 and control4, while original primers targeting H3K27ac regions (EF1 and UBC promoters) are now called control1 and control2. The regions selected are shown in the display above.

In addition, we have tested the effects of both set of primers to normalize H3K27ac signal in the gun1 lincomycin data. The data is shown in the graph below.

As seen in the graph above, the final results do not change with the new normalization (which in this case would be background signal) and the signal in the original control targets is kept constant.

-However, all the ChIP-qPCR for H3K27me3 presented in the manuscript has been exchanged for analysis using the new control primers, control3 and control4 (Figure 2, Figure 3, Figure 4, Supplementary Figure S11). As you can see our main point that H3K27me3 needs to be depleted before the establishment of photosynthesis and that H3K27me3 responds to retrograde signals from the chloroplast remains valid. In fact, your suggestions made the data come out much clearer with a difference now in the H3K27me3 levels in the *gun1* mutant and the *REF6-ox* line (Supplementary Figure S11f). We are grateful for this very specific insight on the underestimation that could stem from using different normalization methods. Nevertheless, our re-analysis demonstrates that our results are solid.

On Figure 4, Figure 7: The set of genes is different than in Figure 3E (ChIP +/- lincomycin in WT), so we do not know if *LHCB2.2*, *GUN4* *PORA*, and *PORB* are affected by the lincomycin in the WT in the first place.

-In Supplementary Figure S12b the effect on Lincomycin on *LHCB2.2*, *GUN4* and *PORB* expression was clearly shown.

The effect of *gun1* on H3K27me3 shown in Fig S11f is missing, and also in Figure 5g where the level of H3K27me3 in *ref6* and *VAL1-ox* is missing.

-After your recommendation for alternate normalization method we now show reduced H3K27me3 in the *gun1* mutant and the *REF6-ox* lines following growth on Lincomycin. The revised figures are shown in Supplementary Figure S11. Thank you for suggesting an alternative and correct method to normalize our ChIP-qPCR.

A statistical test is missing for panel 4a.

-Panel 4a shows the data from two independent ChIP experiments with circles indicating each data point as was described in the legend. We present the ChIP-data as was presented in the publication Jin *et al.*, *LEAFY is a pioneer transcription factor and licenses cell reprogramming to floral fate*. Nature communications, 2021

Line 151: Preferentially cite the original study 10.1105/tpc.13.3.599

-This reference has been included.

Line 180: “H3K27me3 repression was fully lost”, and also in line 397: “complete demethylation” The way the data are presented does not allow for comparing H3K27me3 levels with the background expected and concluding on a “full loss”. Also, as raised in my previous review, it cannot be stated without precaution that H3K27ac gain results in increased PhANG expression as correlation is not causality.

-We apologize and have corrected this overstatement.

Line 178 (old Line 150-151) and Line 419-421: I still don't understand what is the so-called “maintenance mechanism of a closed chromatin state” the authors refer to. And neither how this can explain the “H3K27me3 recover” or “the on-and-off behavior of H3K27me3 deposition”. The authors replied by restating the original manuscript, citing ref 33, which describes the conservation of FIE and PRC2 function in land plants, repressing the differentiation of meristems and does not seem to help the conclusion.

-We have removed this section but there are models that addresses the dynamics of epigenetic nucleosome modification eg. Dodd *et al.*, 2007, Cell, and that cell-to-cell variation does occur and this would be something very interesting to address experimentally in the future using single cell systems.

Figure Supp 6/8: It is stated that meta-profiles are centered around TSS. However, there is a peak of H3 while TSSs are most often devoid of nucleosomes and form so-called nucleosome-free regions. Also, the GLKs peak at both sides of TSS. Is it possible that this is the center of the GLKs' peaks instead of an orientated TSS? In addition, sharp peaks of H3 ChIP-seq at TSS, usually devoid of nucleosomes, in the middle of a 3kb region that should count around ten nucleosomes and not just one is unexpected.

-This is true, and we apologize for our mistake. The figure has been corrected to “Center” as they are regions identified in our histone analysis centered in the middle of the peak.

Figure 5c: Do all the “overlapping” regions have a VAL1 peak? The way similar regions are compared is unclear.

-Supplemental Figure 6 shows that overlap regions have the most VAL1 signal but as shown in Supplemental Figure 8 VAL1 signal is stronger in 2 clusters (up and downstream the Center of peaks, which contain *PhANGs*) and less present in a 3rd cluster.

Line 391: This conclusion is overstated considering that only 12 genes have been tested (fig4).

-We have reformulated.

Line 512-513: A GUN1-RS, active in darkness and under lincomycin, seems rather to prevent acetylation.

-Yes, this is correct.

Supp12: the fact that there is less *PhANG* expression in *val1* is not discussed whereas, if I understood correctly, the opposite effect was expected: VAL1 promoting repressive H3K27me3.

-The *val1* mutant has higher *PhANG* expression in the dark as shown in Figure 6. This expression we think may be driven by the increased expression of *GLK1* observed in the *val1* mutant. In addition, the *VAL-ox* line shows reduced *GLK* expression in the dark. According to our model the main role of VAL1 is in the dark and during the very first hours of light response and at this time point we do see the expected phenotype, increased *PhANG* expression. In Supplementary Figure S12 the seedlings have been exposed to light for 10 hours, thus, here other actors are coming into the picture, for example increased *PhANG* expression can also be detrimental for the seedling as they could be subject to oxidative stress which generates a stress signal that have negative impact on *PhANG* expression. Hence, the observed difference in *PhANG* expression in the *val1* mutant compared to wild type between dark and extended light treatment.

It was previously asked to “analyze in a systematic way, what is the overlap between VAL1 and GLK target genes, do they bind to the same promoters, to the same sites? Is it possible for GLKs to replace VAL1 at promoters?” The authors provide 8 *PhANG* loci and show profiles of the binding of GLK2, VAL1, and REF6 at the “overlap” subset. Unfortunately as shown in the figure Supp7, there is no peak visible for REF6 (except for LHCA2) nor for VAL1, at least using the provided zoom scale. Also, the heatmap in Supp8 and main Figure 5 c and d cannot be interpreted correctly without a positive heatmap of regions known to be enriched in GLK2, VAL1, or REF6 to compare to. Analyzing overlap systematically would have required determining how many genes are found in common between the lists of TF targets (retrieved from the three studies) and the genes in the “overlap” subset.

-In the Supplemental Figure 6 we display that GLK2 is present in profile05 regions and GLK2 and VAL1 are present in the overlap regions. A more detailed, peak by peak analysis can be observed in Supplemental Figure 8 indicating that indeed some regions have dual binding of GLK2 and VAL1. However, this event does not happen at the exact position. Also, the VAL1 ChIP-seq from the published data was performed in 10 day old seedlings where both VAL1 and GLK2 are present in the cell. Hence, a direct replacement seems unlikely.

Last, the following point has been addressed unsatisfactorily by the authors:

“For the greening to be assessed also on cell cultures, in the set up that allowed their identification, and with a proper chlorophyll quantification rather than an RGB green measurements”. The authors replied: “The greening process of the cell culture has been previously described in great detail, e.g. Dubreuil et al., *Plant Physiol.* (2018), Diaz et al., *Nature Comm.* (2018) and Hernandez et al., *New Phytol.* (2022).”

Only chlorophyll measurements supporting the RGB measures have been provided for seedling assays. Yet, to the best of my knowledge, the greening of ref6 or VAL1-OX cell cultures has not been assessed. The involvement of REF6 and VAL1 being inferred from epigenomic data obtained in such an artificial system, it can be important to test the greening of ref6 or VAL1-OX in the same setup.

-There must be a misunderstanding here from the reviewer as there is a pigment quantification shown in Supplementary Figure S10 for *glk1glk2*, *ref6* and *VAL1-ox*. All the mutant analysis shown in this manuscript, Figure 4-7, and Supplementary Figure S9-S12, were performed *in planta* material and not in the “artificial system“ cell culture system. We used the cell culture system as it provides the required temporal resolution to investigate regulatory transitions, we identified interesting histone PTMs and potential candidates for regulatory components. We transferred this information to the seedlings system, where we could confirm the same changes in histone PTMs during the greening, and the involvement of three protein candidates in the process by detailed mutant analysis.

Reviewer #3

The authors have made significant improvements to the manuscript, particularly with the inclusion of the new REF6 data. My previous concerns have been satisfactorily addressed. I now have only a few remaining concerns.

-Thank you for your positive comments!

Major concerns:

1. The genome browser tracks in Supplementary Figure 7 should indicate whether VAL1 or REF6 peaks can be identified with MACS2 or comparable peak caller. The majority of VAL1 and REF6 peaks appear to be background rather than actual peaks.

-Peak calling threshold parameters are arbitrary; thus, we prefer to show qualitative images as such. We agree that in most targets VAL1 appears as background due to the nature of broad domain peaks of this factor for some regions. We have re-addressed the text accordingly. For REF6 we have provided two additional studies where better-defined peaks are presented. We want to point out that VAL1 ChIP-seq available data is performed in adult tissue where we expected VAL1 function to be less prominent, yet we still find some *PhANGs* with VAL1 binding.

2. A better option than REF6 ChIP-seq data could be H3K27me3 data in ref6 mutants since they display more than 1000 genomic regions that ectopically gain H3K27me3. Multiple datasets are already published and might help to further substantiate REF6-mediated regulation of Phang expression.

-The point of Figure S8 was to explore possible overlap in the binding of REF6 and the transition from H3K27me3 to H3K27ac for the *PhANG* genes during the greening process to determine a possible involvement of REF6 in the regulation of these specific genes. What you suggest is a good complement and we have found the key genes in chlorophyll biosynthesis to have both peaks of REF6 and present H3K27me3 hypermethylation in the *ref6* background. They are now displayed in new Supplemental Figure 7b.

Minor concerns:

1. The gene lists in Supplementary Figure 6 need a better explanation. It's unclear which gene sets were used to create the metagene plots.

-For the Metagene profiles of the transcription factors GLK2, VAL1 and REF6 we used published CHIP-seq data, and the references are found in Supplementary Table 1. The specific genes listed are photosynthesis associated genes identified in each of the subsets: Day4 to Day7 Acetylated, Profile 05 and 27me3/27ac Overlap. We have reformulated the legend to Supplementary Figure 6 to be clearer.

2. Do the authors have an explanation for the shift of VAL1 binding from upstream of the TSS to downstream of it in Figure 5C.

-We want to point out that there is no shift in time but some regions that have upstream VAL1 binding and another cluster of VAL1 binding downstream. Actually, Figure 5C is clustered according to VAL1 binding. Nevertheless, it is an interesting observation that we at this point do not have a good explanation for except for the different distribution of VAL1 binding sites.

Detailed response to the Reviewers comments

Reviewer #2

- Thank you for your genuine interest in our work and your very constructive comments along the review process.

1. While claimed as being a major finding of the study (lines 136, 154, 163-164, 319, and the model in Figure 7), evidence for a first phase of H3K27me3 loss preceding a general gain in H3K27ac at RS-regulated genes during seedlings de-etiolation is still lacking.

- In our lincomycin experiments displayed in Figure 3 we are using both Arabidopsis cell culture and seedlings. The data presented clearly demonstrate that H3K27ac deposition is blocked and H3K27me3 deposition maintained following Lincomycin treatment, a condition when the biogenic RS signal is blocked and the GUN1 protein levels are maintained high (Hernández-Verdeja T *et al.*, (2022) GENOMES UNCOUPLED1 plays a key role during the de-etiolation process in Arabidopsis. New Phytologist, 235(1): 188–203). In addition, when using the *gun1* mutant seedlings where this biogenic signal is absent, we display in Figure 4 and supplementary Figure 11b, that H3K27ac deposition is elevated and correlated with high *PhANG* expression, and H3K27me3 deposition reduced compared to wild type, respectively. Furthermore, when we manipulate the receiver in the nucleus of the RS signal, GLK2 we find that GLK2:OX seedlings result in elevated H3K27ac deposition and high *PhANG* expression (Figure 7). Taken together, there is critical and significant data presented in the manuscript providing strong evidence for a first phase of H3K27me3 loss preceding a general gain in H3K27ac at RS-regulated genes during seedlings de-etiolation.

2. The results obtained with seedlings do not reproduce what is shown for the subset of 148 genes in a cultured cell system. In contrast to the claim, the data in Figure 2g and Figure Sup. 3 show either no change or a transient gain of H3K27me3 in response to light, hence if anything an increase but not a decrease. A demonstration would require, for example, a temporal and quantitative analysis of H3K27me3 and H3K27ac absolute levels and not just visual observations at a few genes.

- You are correct that the seedling data is not as clean as the data from the cell culture. The reason for establishing the cell culture was to set up an experimental system where all the cells are homogenous and synchronized. While the single cell culture displays a clear quantitative loss in methylation at the *PhANG* loci during the time course (Figure 1c, 5c Supplemental 6b,8b) it is correct that the seedlings do not show a clear quantitative loss in methylation, however, the seedling result clearly display a loss of H3K27me3 during the retrograde phase (comparing the 12h to 24h time-points). We observed a fast H3K27ac deposition after 6 hours of light but with a decline at 12 hours. At 12 hours light exposure H3K27me3 deposition at the *PhANG* loci is high and H3K27ac low which is symptomatic of the critical checkpoint of the chloroplast development process at this time point (Dubreuil C *et al.*, (2018) Establishment of photosynthesis is controlled by two distinct regulatory phases. Plant Physiology, DOI:10.1104/pp.17.00435). Following 24 hours of light exposure, the chloroplasts have matured, and the retrograde signal has been triggered. At this time point the H3K27me3 mediated repression was reduced while H3K27ac gained, resulting in an increased expression of *PhANGs*. Thus, the retrograde signals play a key role in this final transition from H3K27me3 to H3K27ac during the second phase of chloroplast biogenesis and establishment of photosynthesis. To avoid misinterpretation of the scope of the manuscript we

have readdressed the text to more strongly focus on the message that retrograde signals change the chromatin state of *PhANGs*. We apologize for not correctly formulating ourselves in the original text regarding the seedling data. This has now been corrected.

3. Although this was repeatedly requested, the study still lacks a systematic analysis of the overlap between the genes targeted by VAL1, REF6, and GLKs, and those losing/gaining H3K27me3 and H3K27ac. This hinders the evaluation of the authors' conclusions on the key concept of VAL1, REF6, and GLKs acting in a concerted manner at common specific loci.

- **We appreciate the curiosity to get to the details regarding these three factors identified through our study, however, the key concept of the manuscript is the role of retrograde signals on the chromatin state. We have shown that at specific loci identified from our cell culture that presents differential H3K27ac / H3K27me3 deposition there is a co-occurrence of these factors (using data from published ChIP-seq datasets performed in planta). To be clear, we do not anywhere in the text say that our data demonstrates a mechanistic interplay between VAL1, REF6 and GLK and elucidating such details requires significant additional work and is the scope for another future investigation. From the data we have collected we can say that these factors can be linked to the greening process as mutants and over expressors display impaired H3K27ac deposition that is linked to changes in *PhANG* expression.**

4. Instead, the manuscript displays heatmap signals, meta-profiles across a few selected genes and a few genome browser views (Figure 5c, Sup. 6, Sup. 7, Sup. 8b) that, problematically, unconvincing support the claims of VAL1-REF6-GLK2 co-occurrence at chromatin. More precisely, most GLK2 and REF6 peaks are loosely defined or invisible.

- **We carefully formulate ourselves and do not anywhere in the manuscript claim that VAL1-REF6-GLK2 are the sole regulators of the RS-triggered shift from H3K27me3 to H3K27ac at the specific *PhANG* loci. What we do say is that mutants and transgenic lines of these factors show phenotypes that indicate that they play a role in this process. We do however agree that the REF6 peaks, from published work, are minimal, yet we do find specific *PhANGs* with REF6 regulation of H3K27me3 (Supplemental Figure 7b). While we use published ChIP data from adult plants, which is what is available (very different from de-etiolating cotyledons), we still find strong indications for these factors to be regulating the *PhANGs*. A phenotype which we experimentally confirmed with analysis of mutants and over-expressor lines (Figure 5,6,7, Supplementary Figure 9,10,11, 12).**

5. A demonstration would, for example, be based on correlation tests between the chromatin association of VAL1, GLK2, REF6 and the corresponding histone modifications for each of the VAL1 clusters analyzed, and/or determining how many genes are commonly bound by these proteins upon stringent peak calling (retrieved from the three studies) and the genes in the "overlap" subset.

- **Supplementary Figure 8 displays the overlap between these factors. We agree it may not be quantitative, but on the other hand, comparing ChIP-seq peaks from completely different studies (with different data qualities) could lead to biased correlations (VAL1 broad signal peaks, different peak**

calling thresholds etc.). We identified VAL1, REF6 and GLK2 through our bioinformatic analysis, we then explored and displayed the genomic data as strong indications that lead us to collect and generate mutant and over-expression lines for these factors to experimentally investigate their phenotypes during de-etiolation and RS-mediated changes to histone PTMs. As was formulated in the text, “we address genomic data as a strong indication that leads us to study VAL1, REF6 and GLK2 mutants and over-expressors finding strong physiological indications of their role in photosynthesis establishment as well as their impact in *PhANG* expression and histone deposition in RS conditions”. In our opinion, the described phenotypes of these lines are clearly convincing data suggesting an involvement of VAL1, REF6 and GLK2 in the greening process.

6. Some ChIP-qPCR data are still normalized to a control condition in which we ignore if the tested loci are enriched in the corresponding histone modification or protein. Addressing this point can give rise to drastically different interpretations. Here, in several instances, the control condition corresponds to an unmarked situation (i.e., H3K27ac in darkness in Figures 2a, 2g, Sup. 2, Supp 3 at day 4 and H3K27me3 at day 7 in Figure 3c). Therefore, the ChIP signal of a locus unmarked in the control condition is normalized to background noise and can only give aberrant results. For example, a ChIP signal that decreases as compared to a background signal in the control condition would only quantify noise differences.

- **In the last revision we experimentally addressed the concern of normalization using the normalization method you suggested, with new primer sets from regions constantly marked for the PTM of interest. Our re-analysis demonstrated that our results are solid and the main point of the manuscript; that H3K27me3 needs to be depleted before the final establishment of photosynthesis, and that H3K27Ac deposition responds to retrograde signals from the chloroplast is well supported. As we were unclear in our previous response, we now show detailed information about the control primers that we used and how they “behave”. We have also re-normalized data against conditions when you expect enrichment of the histone PTM, e.g. 7 days in the light for H3K27ac. Please find the information requested below under the specific headings:**

Information about the CONTROL PRIMERS:

H3K27ac controls:

Control1-UBC and Control2-EF1a are constantly marked by H3k27ac (including in Darkness)

While there is no other study than ours evaluating histone PTM under LINCOMYCIN conditions we have strong indications from published RNAseq data that Control1 and Control2 are expressed as house-keeping genes, no change in expression in response to changing conditions, thus it is likely the histone PTM is the same between control and Lincomycin conditions. The RNAseq data is displayed below:

H3K27me3 controls:

Control3 and Control4 are constantly marked by H3K27me3.

The Control3 and Control4 genes are non-expressed genes (not at any point of (de)etiolation or lincomycin treatment) which strongly suggests that Control3 and Control4 are located in regions that are constantly repressed by H3K27me3.

About DATA POINT normalization:

The control primers are shown to be an adequate constant “positive” signal, together with H3 normalization and we have now checked for a possible background effect to address your reasonable concern. For any time-point and condition data point, we first normalized for H3 (which always provides a positive signal of the total histone amounts) and then for the POSITIVE constant histone-PTM signal from the Control regions (which we know to be always marked). Thus, regions of histone-PTM background noise would be identified in our results (regions with background signal would be clearly low, twice divided by positive signals). If any control condition (e.g dark) would behave aberrantly as suggested, the final normalization against this time-point would result in outlining results. We do not see that in our data. Nevertheless, we provide examples normalizing to conditions at time-points where the histone PTM is expected (e.g. we know acetylation is expected in Day7 cells) and obtaining similar results, the new normalized data is shown below:

Normalizing to conditions where histone enrichment is expected (Day 7 acetylation in Phangs) doesn't change the result. Differences between conditions are kept.

Same is observed for H3K27me3 that is expected at Day4

As seen above, irrespective of normalization method, the relative differences are maintained between the conditions examined. We understand you might have a preference to display raw % of Input but we respectfully disagree and prefer to display our data as it stands in the manuscript. We believe we have provided enough evidence with re-running controls and different normalization methods to prove our method valid to detect histone PTM changes by qPCR. It has been reassuring to scrutinise our normalization methods during this review process as you indicate, data normalization can seriously skew the results. We thank you for these insightful and constructive comments on our work which has helped us to feel truly confident that our data is robust.

7. Similarly, in the absence of systematic analysis of H3K27me3 and H3K27ac levels in the starting (dark) and final (10h light) conditions, concluding on general effects in the mutant and overexpressor lines on H3K27ac enrichment by looking at a few selected genes fitting the model in Figure 5g is an over-simplification.

- **The range of genes selected are good representatives of *PhANGs* and their mis-regulation has been proven in numerous studies to be linked to pale phenotypes and impaired photosynthetic function. Additionally, both GLK2 and REF6 overexpression on Lincomycin rescue H3K27ac deposition on these targets, which supports a model of a RS-mediated switch of histone PTM. Taken together with the Lincomycin experiments and all the different mutants and over-expression lines demonstrating clear phenotypes regarding the RS-mediated switch of histone PTM our proposed model is well supported by data.**

8. In contrast to the claim on lane 161, the data do not give information on chromatin opening/closing: this has not been probed at all. Linking H3K27acetylation to chromatin openness is a shortcoming, the latter could be interestingly assessed using histone H3 ChIP-seq or, better, MNase-seq.

- **We have changed the formulation to clearly separate these claims from our study where chromatin accessibility is only described as association to H3K27ac/H3K27me3.**

9. The conclusion on reduced H3 occupancy at central positions of the 'H3K27ac/H3K27me3 overlap cluster' regions or VAL clusters 1-2 (lines 232-6 and Supplementary Figures 6c/d and 8d) is over-interpreted: without quantitative analyses made using the same genome coordinates

- **Supplementary Figure 8 is performed region by region (same genome coordinates) and in panel D we show H3 reduced loss at the very same loci of panel A (each line is a genome coordinate). In all figure panels and precise functional annotations of these regions, the observed lower H3 occupancy could correspond to the formation of nucleosome-free regions linked to increased transcription and H3K27ac gain away from VAL1/GLK2 binding. For example, H3K27ac and H3K27me3 enrichments notoriously occur at promoter/TSS and gene body domains, respectively. We focus on centred peaks histone PTMs which are ranked and clustered by VAL1 signal. H3 loss is not only in cluster 1 and 2 but all "overlap cluster". The loss of H3 in conjunction with a gain in H3K27ac, TF binding and gain in expression (Please see below for a Figure provided of the expression of the overlap genes for clarity) points to a clear gain in accessibility described by many studies, for example, "Comparative analyses show that both H3K27ac and DNase I accessibility highly positively correlate with the expression of nearby genes" (Yan, W., Chen, D., Schumacher, J. *et al.* Dynamic control of enhancer activity drives stage-specific gene expression during flower morphogenesis. *Nat Communications* 10, 1705 (2019)).**

RNA-seq levels of the "overlap cluster" show increased expression.

10. The elegant model builds on direct effects of VAL1 and REF6 on histone H3 acetylation at specific PhAN genes, yet these factors are known to contribute to regulating histone H3 methylation. No experiment supports such the claim of these factors mediating a H3K27me3-to-H3K27ac switch,

- **Regarding the involvement of REF6, Supplementary Figure 11 show the phenotype of the *REF6:OX* line on Lincomycin. The lincomycin experiment is an appropriate read-out for the H3K27me3/H3K27ac switch as it holds the starting condition of chromatin (H3K27me3 high / H3K27ac low). However, in the *REF6:OX* we found lower H3K27me3 and higher H3K27ac compared to wild type. This would suggest an involvement of REF6 in the H3K27me3-to-H3K27ac switch.**

In addition, Figure 5 clearly show reduced H3K27ac in *ref6* mutant and VAL1-OX lines. Figure 6 displays the role of VAL1 and GLK in the regulation of *PhANG* expression and establishment photosynthesis. The *val1* mutant display significant *PhANG* expression in the dark and this is reduced in the *val1glk1glk2* triple mutant demonstrating the genetic connection between VAL1 and GLKs. The effects in the *val1* mutant regarding loss of H3K27me3 deposition have been described previously in detail (Qüesta *et al.*, *Arabidopsis* transcriptional repressor VAL1 triggers Polycomb silencing at *FLC* during vernalization, (2016) *Science*, Vol 353, Issue 6298, pp. 485-488)

11. Especially if considering that H3K27me3 demethylation during seedling de-etiolation is not robustly supported by data and can hardly serve to conclude on a passive and rapid H3K27 acetylation provoked by H3 demethylation at PhANGs. The model should for example invoke and/or test the activity of a histone acetyltransferase.

- **While we have focused on REF6 and VAL1 due to their motif enrichment in our genomic data (both affecting H3K27me3) we show H3K27ac impairment in *glk1glk2* and the opposite in *GLK2ox* in lincomycin, GLKs being a described major driver of de-etiolation.**

There is published work describing a histone deacetylase linked to *PhANG* repression in darkness (Liu, X *et al.*, Associates with the Histone Deacetylase HDA15 in Repression of Chlorophyll Biosynthesis and Photosynthesis in Etiolated *Arabidopsis* Seedlings (2013) *Plant Cell*, Volume 25, Issue 4, Pages 1258–1273). In addition, a similar but yet opposite effect (H3K27ac to H3K27me3 switch) has been demonstrated at the *phyA* locus during dark to light transition (Jang IC *et al.*, Rapid and Reversible Light-Mediated Chromatin Modifications of *Arabidopsis* Phytochrome A Locus (2011) *Plant Cell*, Vol. 23: 459–470).

Reviewer #3

1. The authors satisfyingly revised their manuscript, and I have no further concerns. In response to my first previous concern, I just want to emphasize that identifying peaks through visual inspection using a genome browser is inherently subjective and arbitrary. In contrast, applying thresholds that are widely accepted in the field and established by the ENCODE consortium provides an objective and standardized approach. Therefore, if a peak does not appear even under relaxed threshold settings, it should not be considered a real peak.

- **Thank you for your positive comment and your support. We also acknowledge your insightful advice on peak calling and we will use the approach provided by the ENCODE consortium in the future.**

Reviewer #5

- **OK.**

In the revised manuscript, several critical improvements have been made to the study: the new Figure 4 and S11, now including the gun1 mutant, support the conclusion that a chloroplastic signal controls H3K27 acetylation at photosynthetic genes via GUN1 and the GLK TFs (figure 7 and s12). Interestingly, REF6 also seems necessary for the light-induced increase in H3K27ac at photosynthetic genes (ref6 mutant figure 5) and sufficient to alter the lincomycin-triggered decrease in that mark (REF6-ox figure S11).

Despite these major additions, the mechanism of action of REF6 and VAL1 remains unclear because no effect on H3K27me3 could be reported for those two factors, nor for GUN1. In addition, all major conclusions of the article are drawn from ChIP-qPCR made on a maximum of 8 genes, sometimes genes in different between panels (such as figure 3E), sometimes with only 2 or 3 genes out of 8 showing a difference to the control (figure s10 and s11).

In addition, after the improvement of the manuscript and a better description of the ChIP-qPCR procedure, I now have serious concerns about the ChIP-qPCR data normalization (detailed below), which, presented as it is, calls into question the validity of essential claims.

About ChIP-qPCR data normalization

The ChIP-qPCR graphs (Figure 2 g, Figure 3 e, and f, Figure 4a, Figure 5g, Figure 7a, Supplemental Figures 3, 11e, 11f) result from a series of 4 normalization rounds that can introduce several biases:

1. The ChIP-qPCR values are calculated first, as usually done, as a the percentage of input
2. These values are then normalized against the % of input of an H3 IP on the same genomic target
3. This 2-step normalization is then also applied to two control genes, UBC and EF-1alpha. The geometric mean of the values for these two genes serves then to normalize the previous value (obtained in step 2). **This leads to a first caveat:** as shown by a survey of the ChIP tracks provided with the study for the culture system, **UBC and EF-1alpha are not marked by H3K27me3 but only by H3K27ac in their promoter.** Both primer pairs amplify promoter regions that are unfortunately devoid of H3K27me3. Therefore, I am afraid that all H3K27me3 qPCR data presented in the article is normalized to background noise.

4. Finally, a fourth round of normalization is made towards the value obtained in the control condition. These control conditions are, depending on the figure, the WT dark (2g and supp3), WT light mock (3ef,5g), or WT light Lincomycin (4a,7a, supp11ef). **Here comes the second caveat:**

The 8 genes assessed have been selected for differential marking in the cell culture system, yet proper H3K27ac and H3K27me3 marking have not been verified for each of the three control conditions. If a gene is not marked in the control condition (denominator), then the ratio is a ratio over background noise. This essential point should be checked.

Overall, this 4-step normalization procedure assumes twice that the level of background signal is reproducible between different genome positions, samples, and replicates. As such, there is a reasonable doubt that the observed differences report actual differences in marking.

To avoid biases, qPCR data are usually presented simply as % of input (as such, or relative to the percentage of input of the H3 signal). Moreover, for each histone modification, qPCR figures should display side by side the different time points, culture conditions, or genotypes. Importantly, each qPCR analysis should also include at least one unmarked locus to compare proper signals from background levels, and a marked locus to compare the validity of peak analysis.

Minor comments:

The authors stated in their rebuttal that “H3K27me3 was extremely difficult to monitor during deetiolation as its deposition and removal appear to be slower compared to H3K27ac.” Besides the issue of normalization described above that may explain these difficulties, I wonder if this could be due to differences in cell cycle/mitosis between the two systems. The ten-hour timeframe of seedling de-etiolation is notoriously devoid of cell division and endoreduplication while cultured cells may be dividing fast in both light and dark conditions. It can be hypothesized that there is a passive loss of H3K27me3 due to cell division in the culture system, while this does not happen in cells of de-etiolating seedlings wherein DNA replication is absent or very low.

On Figure 4, Figure 7:

- The set of genes is different than in Figure 3E (ChIP +/- lincomycin in WT), so we do not know if LHCB2.2, GUN4 PORA, and PORB are affected by the lincomycin in the WT in the first place.
- The effect of gun1 on H3K27me3 shown in Fig S11f is missing, and also in Figure 5g where the level of H3K27me3 in ref6 and VAL1-ox is missing.
- A statistical test is missing for panel 4a.

Line 151: Preferentially cite the original study 10.1105/tpc.13.3.599

Line 180: “H3K27me3 repression was fully lost”, and also in line 397: “complete demethylation”

The way the data are presented does not allow for comparing H3K27me3 levels with the background expected and concluding on a “full loss”. Also, as raised in my previous review, it cannot be stated without precaution that H3K27ac gain results in increased PhANG expression as correlation is not causality.

Line 178 (old Line 150-151) and Line 419-421: I still don’t understand what is the so-called “maintenance mechanism of a closed chromatin state” the authors refer to. And neither how this can explain the “H3K27me3 recover” or “the on-and-off behavior of H3K27me3 deposition”. The authors replied by restating the original manuscript, citing ref 33, which describes the conservation of FIE and PRC2 function in land plants, repressing the differentiation of meristems and does not seem to help the conclusion.

Figure Supp 6/8: It is stated that meta-profiles are centered around TSS. However, there is a peak of H3 while TSSs are most often devoid of nucleosomes and form so-called nucleosome-free regions. Also, the GLKs peak at both sides of TSS. Is it possible that this is the center of the GLKs’ peaks instead of an orientated TSS? In addition, sharp peaks of H3 ChIP-seq at TSS, usually devoid of nucleosomes, in the middle of a 3kb region that should count around ten nucleosomes and not just one is unexpected.

Figure 5c: Do all the “overlapping” regions have a VAL1 peak? The way similar regions are compared is unclear.

Line 391: This conclusion is overstated considering that only 12 genes have been tested (fig4)

Line 512-513: A GUN1-RS, active in darkness and under lincomycin, seems rather to prevent acetylation.

Supp12: the fact that there is less PhANG expression in val1 is not discussed whereas, if I understood correctly, the opposite effect was expected: VAL1 promoting repressive H3K27me3.

Points not addressed in the rebuttal:

It was previously asked to “analyze in a systematic way, what is the overlap between VAL1 and GLK target genes, do they bind to the same promoters, to the same sites? Is it possible for GLKs to replace VAL1 at promoters?”

The authors provide 8 PhANG loci and show profiles of the binding of GLK2, VAL1, and REF6 at the “overlap” subset.

Unfortunately as shown in the figure Supp7, there is no peak visible for REF6 (except for LHCA2) nor for VAL1, at least using the provided zoom scale.

Also, the heatmap in Supp8 and main Figure 5 c and d cannot be interpreted correctly without a positive heatmap of regions known to be enriched in GLK2, VAL1, or REF6 to compare to. Analyzing overlap systematically would have required determining how many genes are found in common between the lists of TF targets (retrieved from the three studies) and the genes in the “overlap” subset.

Last, the following point has been addressed unsatisfactorily by the authors:

“For the greening to be assessed also on cell cultures, in the set up that allowed their identification, and with a proper chlorophyll quantification rather than an RGB green measurements”.

The authors replied: “The greening process of the cell culture has been previously described in great detail, e.g. Dubreuil et al., *Plant Physiol.* (2018), Diaz et al., *Nature Comm.* (2018) and Hernandez et al., *New Phytol.* (2022).”

Only chlorophyll measurements supporting the RGB measures have been provided for seedling assays. Yet, to the best of my knowledge, the greening of ref6 or VAL1-OX cell cultures has not been assessed. The involvement of REF6 and VAL1 being inferred from epigenomic data obtained in such an artificial system, it can be important to test the greening of ref6 or VAL1-OX in the same setup.